# AN INFORMATION FUSION APPROACH TO LEARNING WITH INSTANCE-DEPENDENT LABEL NOISE

**Zhimeng Jiang[1], Kaixiong Zhou[2], Zirui Liu[2], Li Li[3], Rui Chen[3], Soo-Hyun Choi[4],[\*] Xia Hu[2]**
[1]Texas A&M University, [2]Rice University, [3]Samsung Research America, [4]Samsung Electronics

## ABSTRACT

Instance-dependent label noise (IDN) widely exists in real-world datasets and usually misleads the training of deep neural networks. Noise transition matrix (NTM) (i.e., the probability that clean labels flip into noisy labels) is used to characterize the label noise and can be adopted to bridge the gap between clean and noisy underlying data distributions. However, most instances are long-tail, i.e., the number of occurrences of each instance is usually limited, which leads to the gap between the underlying distribution and the empirical distribution. Therefore, the genuine problem caused by IDN is *empirical*, instead of underlying, *data distribution mismatch* during training. To directly tackle the empirical distribution mismatch problem, we propose *posterior transition matrix* (PTM) to posteriorly model label noise given limited observed noisy labels, which achieves *statistically consistent classifiers*. Note that even if an instance is corrupted by the same NTM, the intrinsic randomness incurs different noisy labels, and thus requires different correction methods. Motivated by this observation, we propose an **I**nformation **F**usion (IF) approach to fine-tune the NTM based on the estimated PTM. Specifically, we adopt the noisy labels and model predicted probabilities to estimate the PTM and then correct the NTM in *forward propagation*. Empirical evaluations on synthetic and real-world datasets demonstrate that our method is superior to the state-of-the-art approaches, and achieves more stable training for instance-dependent label noise.

## 1 INTRODUCTION

Data labels annotated from human efforts, such as crowdsourcing (Yan et al., 2014; Chen et al., 2017) and online queries (Divvala et al., 2014), may be heavily noisy in practice (Wei et al., 2022). To make it worse, the label noise stemmed from human annotations is often instance-dependent. For example, the images close to the decision boundary are usually prone to be mislabeled (Zhang et al., 2021b; Zhu et al., 2021b). On the other hand, the remarkable success of deep neural networks (DNNs) on supervised learning tasks heavily relies on the expressive power and a large number of data with accurate labels. Unfortunately, deep neural networks memorizes noisy labels leading to poor generalization (Zhang et al., 2017). It is challenging to learn with practical instance-dependent label noise (IDN) due to the hidden and complicated label noise properties (Liu, 2021; Zhu et al., 2022b; Cheng et al., 2021b; Zhu et al., 2022a).

The methods dealing with noisy labels fall into two lines, including heuristically identifying noisy samples and statistical label noise modeling. The training of deep neural networks often learns clean labels first (Arpit et al., 2017) and then gradually memorizes noisy labels, which is recognized as the memorization effect. Based on the general memorization effect, the heuristic methods are all designed by following the anomaly detection strategy: identify noisy samples based on different behaviors (e.g., loss values) between clean and noisy samples during training Cheng et al. (2021a). The typical methods contain sample selection (Yu et al., 2019; Han et al., 2018b), reweight samples (Cheng et al., 2021a; Jiang et al., 2018; Ren et al., 2018), label correction (Ma et al., 2018; Tanaka et al., 2018), and regularization (Han et al., 2018a). Although these algorithms empirically work well, the reliability cannot be guaranteed explicitly without modeling label noise.

Another line of works relies on *noise transition matrix* (NTM) to model label noise statistically (Xia et al., 2019; 2020; Patrini et al., 2017) by quantifying the probabilities that clean labels flip into noisy

---

*Corresponding author (soohyunc@gmail.com)

labels. Although the NTM-based methods possess theoretical guarantee, NTM estimation for each instance under IDN is pretty challenging. To ease the estimation, some unrealistic assumptions have been made on NTM, including instance-independent transition matrix (Liu & Guo, 2020; Wei & Liu, 2021; Li et al., 2021), symmetric transition matrix (Menon et al., 2018), upper bounded noise rate (Cheng et al., 2020), and part-dependent label noise (Xia et al., 2020). However, under the complex IDN, the empirical noise distribution could be highly different from the underlying noise distribution. For example, in Figure 1, the underlying and empirical noisy distributions for long-tail instances are different since the empirical noisy label can be either the same as or different from the clean label. Additionally, observed noisy labels provide *inductive bias* toward label corruption. In other words, the genuine problem arising from IDN is the *empirical*, instead of underlying, clean and noisy distribution mismatch problem.

To mitigate the empirical distribution mismatch problem, we propose the posterior transition matrix (PTM) to model label noise given the observed noisy labels. We adopt PTM to provably bridge the gap between clean and noisy underlying data distributions, and empirical distribution of all anchor points (i.e., data points that belong to a specific class almost surely (Xia et al., 2020)) simultaneously. We also provide an easy-to-compute PTM estimation method under the low label noise condition. To further extend the applicability, motivated by Kalman filtering (Kalman, 1960), we propose the information fusion (IF) method to linearly combine the estimated NTM and PTM, which achieves lower transition matrix estimation error. We empirically show that the proposed IF method can achieve higher accuracy and more stable training. The main contributions of this work are summarized below.

- We propose a new concept of PTM to achieve statistically consistent classifiers for underlying distribution mismatch and anchor point empirical distribution mismatch simultaneously.
- We propose a simple yet effective PTM estimation method based on observed noisy labels under the condition of low label noise ratio. To extend the applicability, we propose an IF method, which combines the estimated NTM and PTM to achieve lower estimation error with both theoretical and experimental justifications.
- We experimentally demonstrate that the proposed IF method achieves higher accuracy and more stable training under synthetic and real-world label noise.

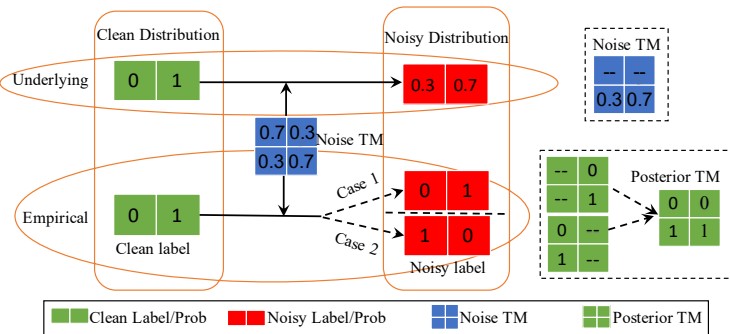

Figure 1: Illustration of the thought experiment. Assume we use *identified* noise transition matrix (NTM) to corrupt the instance with clean label $[0, 1]$. From the underlying distribution perspective, the noisy underlying distribution is $[0.3, 0.7]$. From the empirical distribution perspective, the label of this sample is *statistically* either the same as (case 1) or different from (case 2) the clean label, leading to different PTM. In other words, the observed noisy labels (posterior information) provide *inductive bias* of label correction, which motivates the notion of PTM.

## 2 PRELIMINARIES

Throughout this paper, we adopt uppercase letters to denote random variables, and lowercase letters to denote particular realization of the random variables. We target the $c$-class classification problem with instance-dependent label noise (IDN). Let $\mathbb{P}(X, Y)$ be the *underlying clean distribution* of the random variables $(X, Y)$, where $X$ and $Y$ represent the instance and clean label, respectively. In real-world scenarios, clean labels are usually corrupted, and thus only noisy labels, denoted as $\tilde{Y}$, are observed. Given a set of $N$ training instances denoted by $\tilde{D} := \{(\boldsymbol{x}_n, \tilde{y}_n)\}_{n=1}^N$, where $\boldsymbol{x}_n$ is the

instance vector of the $n$-th sample and $\tilde{y}_n \in [c] := \{1, \cdots, c\}$ is the corresponding observed noisy label, our goal is to predict the clean label $y_n$ for any given instance $\boldsymbol{x}_n$.

**Data distribution.** We assume that the unobserved clean samples $(\boldsymbol{x}_n, y_n)$ and available noisy samples $(\boldsymbol{x}_n, \tilde{y}_n)$ are drawn from the unknown *underlying clean distribution* $\mathbb{P}(X, Y)$ and the *underlying noisy distribution* $\mathbb{P}(X, \tilde{Y})$, respectively. For the noisy samples $\tilde{D}$, we may approximate the underlying noisy distribution by the *empirical noisy distribution*, i.e., $\hat{\mathbb{P}}_{\tilde{D}}(X, \tilde{Y}) = \frac{1}{N} \sum_{n=1}^{N} \delta(X = \boldsymbol{x}_n, \tilde{Y} = \tilde{y}_n)$, where $\delta(X = \boldsymbol{x}_n, \tilde{Y} = \tilde{y}_n)$ is a Dirac mass centered at $(\boldsymbol{x}_n, \tilde{y}_n)$. Since clean data samples are unobserved, we define the posterior empirical clean distribution $\hat{\mathbb{P}}(X, Y | \tilde{D})$ as the inferred posterior empirical clean distribution from the noisy samples $\tilde{D}$. Based on Bayes' rule, the posterior empirical clean distribution is given by $\hat{\mathbb{P}}(X, Y | \tilde{D}) = \frac{1}{N} \sum_{n=1}^{N} \sum_{y_n=1}^{c} \delta(X = \boldsymbol{x}_n, Y = \tilde{y}_n) \mathbb{P}(Y = y_n | \tilde{Y} = \tilde{y}_n, X = \boldsymbol{x}_n)$.

**Noise transition matrix $T(\boldsymbol{x})$.** To describe the corruption process of the clean label, the NTM $T(\boldsymbol{x}) \in \mathbb{R}^{c \times c}$ is defined as $T_{i,j}(\boldsymbol{x}) = \mathbb{P}(\tilde{Y} = j | Y = i, X = \boldsymbol{x})$, which represents the transition probability of flipping the instance $\boldsymbol{x}$'s label from the clean $i$-th class to the noisy $j$-th class. The probability of the underlying noisy label $\tilde{Y}$ given the instance $\boldsymbol{x}$ satisfies:

$$\mathbb{P}(\tilde{Y} = j | X = \boldsymbol{x}) = \sum_{i=1}^{c} \mathbb{P}(\tilde{Y} = j | Y = i, X = \boldsymbol{x}) \mathbb{P}(Y = i | X = \boldsymbol{x}) = \sum_{i=1}^{c} T_{ij}(\boldsymbol{x}) \mathbb{P}(Y = i | X = \boldsymbol{x}).$$

The NTM $T(\boldsymbol{x})$ hence bridges the gap between the underlying clean label probability $\mathbb{P}(\mathbf{Y} | X = \boldsymbol{x}) = \left[ \mathbb{P}(Y = 1 | X = \boldsymbol{x}), \cdots, \mathbb{P}(Y = c | X = \boldsymbol{x}) \right]^{\top}$ and underlying noisy label probability $\mathbb{P}(\tilde{\mathbf{Y}} | X = \boldsymbol{x}) = \left[ \mathbb{P}(\tilde{Y} = 1 | X = \boldsymbol{x}), \cdots, \mathbb{P}(\tilde{Y} = c | X = \boldsymbol{x}) \right]^{\top}$ given the instance $\boldsymbol{x}$, i.e., $\mathbb{P}(\tilde{\mathbf{Y}} | X = \boldsymbol{x}) = T(\boldsymbol{x})^{\top} \mathbb{P}(\mathbf{Y} | X = \boldsymbol{x})$.

**Loss correction method.** Let function $f(\cdot)$ represent a neural network and $f(\boldsymbol{x})$ denote the $c$-dimensional output probability for instance $\boldsymbol{x}$, where the $i^{th}$ index of the output $f_i(\boldsymbol{x})$ represents the predicted probability for class $i$. The common approach is to minimize the cross-entropy (CE) loss $l(f(\boldsymbol{x}), y) := -\log(f_y(\boldsymbol{x}))$ to force the output $f_y(\boldsymbol{x})$ to approximate 1. However, the label noise may mislead a deep learning model. The existing NTM-based methods first estimate NTM $T(\boldsymbol{x})$ and then adopt it to correct the loss function. For example, in the forward correction procedure (Patrini et al., 2017), the estimated NTM is adopted to corrupt the predicted probability $f(\boldsymbol{x})$, i.e., the corrupted predicted probability is $\tilde{f}(\boldsymbol{x}) = T(\boldsymbol{x})^{\top} f(\boldsymbol{x})$, and then the corrupted predicted probability is enforced to approximate the noisy label $\tilde{y}$. Suppose $T(\boldsymbol{x})$ is non-singular and the loss function is proper and composite. The *forward loss correction* can achieve a *consistent classifier*, i.e., the optimal classifier for the corrected loss with respect to the underlying noisy distribution is the same as that for the CE loss with respect to the underlying clean distribution:

$$\arg \min_f \mathbb{E}_{(X, \tilde{Y}) \sim \mathbb{P}(X, \tilde{Y})} [l(\tilde{Y}, T(X)^{\top} f(X))] = \arg \min_f \mathbb{E}_{(X, Y) \sim \mathbb{P}(X, Y)} [l(Y, f(X))]. \tag{1}$$

## 3 LEARNING WITH INFORMATION FUSION

Even though the existing loss correction methods could achieve consistent classifiers theoretically, their performances are still undesirable in practice. Since a deep learning model is often trained on the empirical distribution with limited samples, the NTM cannot correctly bridge the empirical clean distribution and empirical noisy distribution, i.e., $\hat{\mathbb{P}}(\tilde{Y} = j | X = \boldsymbol{x}) \neq \sum_{i=1}^{c} T_{ij}(\boldsymbol{x}) \hat{\mathbb{P}}_{\tilde{D}}(Y = i | X = \boldsymbol{x})$. In other words, the NTM-based method may not work well in practice due to the empirical clean and noisy distribution mismatch problem in IDN.

With the goal of utilizing the observed noisy labels, we propose a concept, named posterior transition matrix (PTM), to describe the transition probabilities given the observed noisy labels (formally defined in Section 3.1). The motivation for PTM stems from a simple thought experiment as shown in Figure 1, which shows that PTM can model the empirical label noise better than NTM. Figure 2 illustrates the

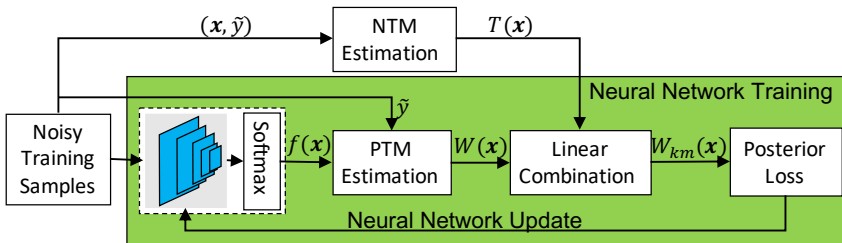

Figure 2: The overview of IF. NTM can be estimated beforehand based on the previous work (Xia et al., 2020). PTM estimation is iteratively obtained based on model prediction and observed label. Subsequently, IF adopts a linear combination for NTM and PTM to reduce the estimation error. Finally, a posterior loss function is proposed to tackle IDN.

overall framework of IF, including NTM and PTM estimations (Section 3.2), information fusion via linear combination (Section 3.3), and posterior loss function (Section 3.1).

## 3.1 THE LOSS CORRECTION METHOD

The main goal is to train a $c$-class neural network classifier $f(\boldsymbol{x}, \omega)$ to predict the clean label probability $\mathbb{P}(Y|X)$. Since only the noisy labels are observed, there is a gap between the clean and noisy label, described via NTM (Goldberger & Ben-Reuven, 2017).

Motivated by the observed noisy labels (i.e., posterior information), we define the PTM $W(\boldsymbol{x})$ to describe the posterior clean label probability given noisy labels, where $W_{i,j}(\boldsymbol{x}) = \mathbb{P}(Y = i|\tilde{Y} = j, X = \boldsymbol{x})$. We provide the relationship between the PTM $W(\boldsymbol{x})$ and NTM $T(\boldsymbol{x})$ via Bayes' rule:

$$W_{i,j}(\boldsymbol{x}) = \frac{\mathbb{P}(Y = i, \tilde{Y} = j|X = \boldsymbol{x})}{\mathbb{P}(\tilde{Y} = j|X = \boldsymbol{x})} = \frac{\mathbb{P}(Y = i|X = \boldsymbol{x})T_{ij}(\boldsymbol{x})}{\sum_{i=1}^{c} \mathbb{P}(Y = i|X = \boldsymbol{x})T_{ij}(\boldsymbol{x})}. \tag{2}$$

Notice that the summation of any column is 1 for PTM $W(\boldsymbol{x})$, while the summation of any row is 1 for NTM $T(\boldsymbol{x})$. Subsequently, we provide a posterior reweight loss correction method via NTM.

**Definition 1 (Posterior reweight loss)** *Assume the model prediction is $f(\boldsymbol{x})$ for noisy sample $(\boldsymbol{x}, \tilde{y})$ and $W(\boldsymbol{x})$ is the PTM associated with the noisy sample. The posterior reweight loss is defined as*

$$l_{p-rew}\Big(\tilde{y}, f(\boldsymbol{x})\Big) = \sum_{i=1}^{c} W_{i,\tilde{y}}(\boldsymbol{x})l(i, f(\boldsymbol{x})). \tag{3}$$

We next analyze the property of the posterior reweight loss and provide the theoretical justification. Specifically, we analyze the expected risk $R_{\mathbb{P}(X,\tilde{Y})}(f) = \mathbb{E}_{(\boldsymbol{x},\tilde{y})\sim\mathbb{P}(X,\tilde{Y})}[l_{p-rew}\Big(\tilde{y}, f(\boldsymbol{x})\Big)]$ and empirical risk $\hat{R}_{\hat{\mathcal{D}}}(f) = \frac{1}{N}\sum_{n=1}^{N} l_{p-rew}\Big(\tilde{y}_n, f(\boldsymbol{x}_n)\Big)$ under the noisy samples.

**Theorem 3.1 (Statistically Consistent Classifier)** *The posterior reweight loss can achieve an consistent classifier for the underlying distribution and empirical distribution.*

*(i) For the underlying distribution, the expected risk satisfies*

$$\arg\min_{f} \mathbb{E}_{(\boldsymbol{x},\tilde{y})\sim\mathbb{P}(X,\tilde{Y})}\Big[l_{p-rew}\big(\tilde{y}, f(\boldsymbol{x})\big)\Big] = \arg\min_{f} \mathbb{E}_{(\boldsymbol{x},y)\sim\mathbb{P}(X,Y)}[l(y, f(\boldsymbol{x}))]. \tag{4}$$

*(ii) For the anchor point samples $(\boldsymbol{x}_{ap}, \tilde{y}_{ap})$ with the underlying clean probability $\mathbb{P}(Y = y_{ap}|X = \boldsymbol{x}_{ap})$, the empirical risk satisfies $l_{p-rew}(\tilde{y}_{ap}, f(\boldsymbol{x}_{ap})) = l(y_{ap}, f(\boldsymbol{x}_{ap}))$. For the empirical distribution, the empirical risk satisfies*

$$\arg\min_{f} \mathbb{E}_{(\boldsymbol{x},\tilde{y})\sim\hat{\mathbb{P}}_{\tilde{D}}(X,\tilde{Y})}\Big[l_{p-rew}\big(\tilde{y}_n, f(\boldsymbol{x}_n)\big)\Big] = \arg\min_{f} \mathbb{E}_{(\boldsymbol{x},y)\sim\hat{\mathbb{P}}(X,Y|\tilde{D})}\Big[l(y_n, f(\boldsymbol{x}_n))\Big]. \tag{5}$$

We also show that the PTM can be adopted in the forward manner. The definition of the posterior reweight loss is given below.

**Definition 2 (Posterior forward loss)** *Assume the model prediction is $f(\boldsymbol{x})$ for noisy sample $(\boldsymbol{x}, \tilde{y})$ and $W(\boldsymbol{x})$ is the PTM associated with the noisy sample. The posterior forward loss is defined as*

$$l_{p-fw}\big(\tilde{y}, f(\boldsymbol{x})\big) = l\Big(\tilde{y}, \sum_{i=1}^{c} W_{i,\tilde{y}}(\boldsymbol{x}) f_i(\boldsymbol{x})\Big). \tag{6}$$

We next analyze the property of the posterior forward loss and provide the theoretical justification.

**Lemma 3.2** *The posterior forward loss is not larger than the posterior reweight loss* $l_{p-rew}\big(\tilde{y}, f(\boldsymbol{x})\big) \geq l_{p-fw}\big(\tilde{y}, f(\boldsymbol{x})\big)$.

**Theorem 3.3 (Statistically Consistent Classifier)** *Given that the loss function $l(y, f)$ is convex with respect to $f$ (the convex condition can be commonly satisfied, e.g., the cross-entropy loss) and that the minimum expected risk $R_{\mathbb{P}(X,Y)}(f)$ can achieve $0$, the posterior forward loss can achieve an consistent classifier for the underlying distribution,*

$$\arg\min_{f} \mathbb{E}_{(\boldsymbol{x},\tilde{y}) \sim \mathbb{P}(X,\tilde{Y})}\Big[l_{p-fw}\big(\tilde{y}, f(\boldsymbol{x})\big)\Big] = \arg\min_{f} \mathbb{E}_{(\boldsymbol{x},y) \sim \mathbb{P}(X,Y)}[l(y, f(\boldsymbol{x}))]. \tag{7}$$

By comparing the posterior forward loss with the forward loss (Patrini et al., 2017), it can be seen that both of them correct the output of a neural network with statistically consistent risk guarantee, while the difference falls into the different transition matrices used in the loss correction. In addition, the statistically consistent risks for the forward loss and posterior forward loss both require accurate transition matrix estimation. However, the NTM and PTM estimation is highly challenging since label noise is instance-dependent in reality. Such an observation motivates us to combine both NTM $T(\boldsymbol{x})$ and PTM $W(\boldsymbol{x})$ to correct the loss.

Motivated by Kalman filtering (Kalman, 1960), a famous estimation method combining prior knowledge and measurement information, we propose an Information Fusion (IF) approach to tackle instance-dependent label noise via loss correction. Specifically, we first claim that there is prior knowledge information and measurement information in the label noise, where prior knowledge information and measurement information correspond to the estimated NTM and PTM, respectively.

## 3.2 PTM Estimation

The posterior loss correction requires knowing PTM $W(\boldsymbol{x})$ given any instance $\boldsymbol{x}$. However, the PTM is unknown and needs to be estimated. In this subsection, we provide a simple yet effective PTM estimation method under the *condition* that the neural network output $f(X)$ probability can well approximate the underlying probability $\mathbb{P}(Y|X)$, i.e., $f(\boldsymbol{x}) \approx W(X = \boldsymbol{x})\hat{\mathbb{P}}(\tilde{\mathbf{Y}}|X = \boldsymbol{x})$. Intuitively, this condition seems strong since

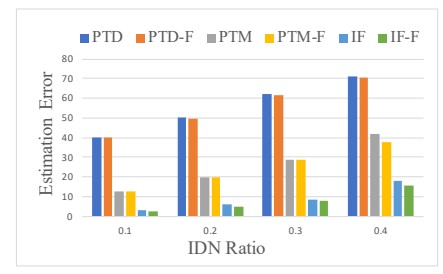

Figure 3: Estimation error for CIFAR10

a neural network is prone to overfit noisy data. To achieve this condition, the warm-up training strategy and iterative PTM estimation are adopted. Warm-up training can make "good" neural network outputs approximating the empirical clean distribution since the network fits clean samples in the beginning of training (Liu et al., 2020). Subsequently, we iteratively estimate the PTM for loss correction during training.

For the case that the number of occurrences of instance $\boldsymbol{x}$ is large, the empirical noisy distribution converges to the underlying noisy distribution, i.e., $\hat{\mathbb{P}}(\tilde{\mathbf{Y}}|X = \boldsymbol{x}) \longrightarrow \mathbb{P}(\tilde{\mathbf{Y}}|X = \boldsymbol{x})$. For a long-tail instance with only a single occurrence $(\boldsymbol{x}, \tilde{y})$, the empirical noisy label distribution is always one-hot. Thus, this condition requires that the output of neural network $f(\boldsymbol{x})$ should approximate to the underlying posterior clean probability, which is quite strong.

Under this condition, we propose a PTM estimation method based on the noisy labels and the model predicted probability. Compared with the NTM estimation method (Xia et al., 2019; 2020), the proposed estimation method does not require the anchor point assumption that some data points

belong to a specific class almost surely. Specifically, the PTM can be estimated as $\hat{W}_{i,j}(\boldsymbol{x}) = f_i(\boldsymbol{x})$ if $\tilde{y} = j$, otherwise is 0. For the case that the number of occurrences is larger than one, the PTM estimation is ill-posed since the number of unknown variables $c^2$ is larger than the number of constraint $c$ (i.e., number of classes). Thus, we choose the PTM with the minimum Frobenius norm, which is consistent with the case of having only a single occurrence. Specifically, we have the following theorem.

**Theorem 3.4 (Posterior transition matrix estimation)** *The optimal solution to finding the PTM with the minimal Frobenius norm, i.e.,*

$$\hat{W}(\boldsymbol{x}) = \arg\min_{W} ||W||_F^2$$

$$s.t. W^\top(\boldsymbol{x})\hat{\mathbb{P}}(\tilde{Y}|\boldsymbol{x}) = f(\boldsymbol{x}), \text{ for } \forall \boldsymbol{x}$$

*is given below:*

$$\hat{W}(\boldsymbol{x}) = \frac{f(\boldsymbol{x})\hat{\mathbb{P}}(\tilde{Y}|\boldsymbol{x})^\top}{||\hat{\mathbb{P}}(\tilde{Y}|\boldsymbol{x})||_2^2}. \tag{8}$$

Theorem 3.4 provides the PTM estimation for general empirical noisy label distributions. For the case of instance $\boldsymbol{x}$ with only a single occurrence, the empirical noisy distribution satisfies $||\hat{\mathbb{P}}(\tilde{Y}|\boldsymbol{x})||_2 = 1$, and achieves consistent estimated PTM in Equation (8).

### 3.3 INFORMATION FUSION

Section 3.2 introduces the PTM estimation method based on the observed noisy labels. However, the condition that a neural network approximates clean labels could still be strong even after the warm-up strategy and iterative estimation are adopted, and PTM estimation error could be large for large IDN rates. To further reduce the estimation error, motivated by Kalman filtering, we propose the information fusion (IF) approach to obtain more accurate transition matrix estimation via weighted average of PTM estimation and NTM estimation with both intuitive and theoretical justifications[1]. Intuitively, for each instance, the estimated NTM and PTM may have different estimation accuracy, and, therefore, it is possible to obtain a more accurate transition matrix estimation by adaptively and linearly combine these two matrices. Specifically, we first quantify the estimation uncertainty and assign higher weight for the estimation with lower uncertainty. In this way, a more accurate estimated transition matrix can be generated.

Before illustrating the information fusion algorithm, we need to quantify the transition matrix estimation error without ground truth. Suppose posterior forward loss is adopted. For the estimated NTM $\hat{T}(\boldsymbol{x})$, the corrupted model predicted probability is $\tilde{f}(\boldsymbol{x}) = \hat{T}^\top(\boldsymbol{x})f(\boldsymbol{x})$. If the corrupted model predicted probability is accurate, the noisy label $\tilde{Y}$ satisfies $c$-dimension Bernoulli distribution with parameter $\tilde{f}(\boldsymbol{x})$, i.e., $\tilde{Y} \sim Bernoulli(\hat{T}^\top(\boldsymbol{x})f(\boldsymbol{x}))$, where $Bernoulli(\cdot)$ represents a multi-dimension Bernoulli distribution. Furthermore, we define the uncertainty $\sigma_{\hat{T}}$ for transition matrix $\hat{T}(\boldsymbol{x})$ as the trace of the covariance matrix for $\tilde{Y}$, i.e., $\sigma_{\hat{T}}(\boldsymbol{x}) = Tr(K_{\tilde{Y}}(\boldsymbol{x}))$, where $Tr(\cdot)$ means the trace of a matrix, $K_{\tilde{Y}}(\boldsymbol{x})$ represents the covariance matrix of random variable $\tilde{Y}$. For the multi-dimension Bernoulli distribution, we have the covariance matrix as follows,

$$K_{\tilde{Y}} = \mathbb{E}[(\tilde{Y} - \mathbb{E}[\tilde{Y}])(\tilde{Y} - \mathbb{E}[\tilde{Y}])^\top] = \text{diag}(\hat{T}^\top(\boldsymbol{x})f(\boldsymbol{x})) - \hat{T}^\top(\boldsymbol{x})f(\boldsymbol{x})f(\boldsymbol{x})^\top\hat{T}(\boldsymbol{x}).$$

Taking trace operation on the covariance matrix, we have the uncertainty for NTM as $\sigma_{\hat{T}}(\boldsymbol{x}) = ||\hat{T}^\top(\boldsymbol{x})f(\boldsymbol{x})||_1 - ||\hat{T}^\top(\boldsymbol{x})f(\boldsymbol{x})||_2$. Similarly, the uncertainty for the PTM is given by $\sigma_{\hat{W}}(\boldsymbol{x}) = ||\hat{W}^\top(\boldsymbol{x})f(\boldsymbol{x})||_1 - ||\hat{W}^\top(\boldsymbol{x})f(\boldsymbol{x})||_2$. Once the uncertainty has been established, we further integrate the two estimated transition matrices into a Kalman transition matrix, defined as $W_{km}(\boldsymbol{x})$, via a weighted average operation. Mathematically, the Kalman transition matrix is given by

$$W_{km}(\boldsymbol{x}) = (1 - \lambda(\boldsymbol{x}))\hat{T}(\boldsymbol{x}) + \lambda(\boldsymbol{x})\hat{W}(\boldsymbol{x}), \tag{9}$$

---

[1]The core idea for more accurate estimation is similar to statistical efficiency (Gong et al., 2020) via generating two "auxiliary" estimators. The key difference is the generation manner: IF generates NTM and PTM from the prior and posterior perspectives, while statistical efficiency generates two different noisy labels.

where the Kalman gain $\lambda(\boldsymbol{x})$ is carefully selected to minimize the estimation error. Let the true PTM be $W^*(\boldsymbol{x})$, where $W_{ij}^*(\boldsymbol{x}) = 1$ if $y = i$ and $\tilde{y} = j$, otherwise 0. Subsequently, we define the reconstruction error for NTM and PTM as $e_T(\boldsymbol{x}) = W^*(\boldsymbol{x}) - \hat{T}(\boldsymbol{x})$ [2] and $e_W(\boldsymbol{x}) = W^*(\boldsymbol{x}) - \hat{W}(\boldsymbol{x})$, respectively. The parameter $\lambda(\boldsymbol{x})$ is determined via minimizing the mean square reconstruction error. We theoretically justify the superiority of IF for a general scenario.

**Theorem 3.5 (Theoretical justification of IF)** *Let the correlation coefficient between reconstruction error $e_T(\boldsymbol{x})$ and $e_W(\boldsymbol{x})$ be $cov(\boldsymbol{x}) \in [-1, 1]$. The trace of the covariance matrix for error $e_{\hat{T}}(\boldsymbol{x})$ and $e_{\hat{W}}$ can be quantified by $\sigma_{\hat{T}}(\boldsymbol{x})$ and $\sigma_{\hat{W}}(\boldsymbol{x})$, i.e., $Tr(e_{\hat{T}}(\boldsymbol{x})e_{\hat{T}}^\top(\boldsymbol{x})) = \sigma_{\hat{T}}(\boldsymbol{x})$ and $Tr(e_{\hat{W}}(\boldsymbol{x})e_{\hat{W}}^\top(\boldsymbol{x})) = \sigma_{\hat{W}}(\boldsymbol{x})$. For the Kalman transition matrix $W_{km}$, let the reconstruction error be $\mathcal{L}_{mse}(\lambda(\boldsymbol{x})) = Tr\Big((W^*(\boldsymbol{x}) - W_{km}(\boldsymbol{x}))(W^*(\boldsymbol{x}) - W_{km}(\boldsymbol{x}))^\top\Big)$. Then the optimal Kalman gain to minimize the reconstruction error is given by*

$$\lambda^*(\boldsymbol{x}) \quad = \quad \arg\min_\lambda \mathcal{L}_{mse}(\lambda) = \frac{\sigma_{\hat{T}}(\boldsymbol{x}) - cov(\boldsymbol{x})\sqrt{\sigma_{\hat{T}}(\boldsymbol{x})\sigma_{\hat{W}}(\boldsymbol{x})}}{\sigma_{\hat{T}}(\boldsymbol{x}) + \sigma_{\hat{W}}(\boldsymbol{x}) - 2cov(\boldsymbol{x})\sqrt{\sigma_{\hat{T}}(\boldsymbol{x})\sigma_{\hat{W}}(\boldsymbol{x})}}, \quad (10)$$

*and the minimum reconstruction error satisfies $\mathcal{L}_{mse}(\lambda^*(\boldsymbol{x})) \leq \min(\sigma_{\hat{T}}(\boldsymbol{x}), \sigma_{\hat{W}}(\boldsymbol{x}))$.*

Theorem 3.5 implies that the Kalman transition matrix $W_{km}(\boldsymbol{x})$ provably achieves lower estimation error for general $e_T(x)$ and $e_W(x)$, either correlated or independent.

## 4 EXPERIMENTS

In this section, we empirically evaluate the effectiveness and stability of our IF approach on synthetic and real-world noisy datasets. We aim to answer two questions as follows. **Q1**: Compared with the state-of-the-art transition matrix based methods, can IF achieve higher accuracy? **Q2**: Can IF lead to more accurate and robust transition matrix estimation and more stable training?

### 4.1 EXPERIMENTAL SETTINGS

**Datasets.** We verify the superiority of IF on three manually corrupted datasets, i.e., F-MNIST, SVHN, CIFAR-10, and one real-world noisy dataset Clothing1M. The first three datasets contain clean data, and we manually corrupt the labels of the training datasets by following (Xia et al., 2020). IDN-$\tau$ means that the controlled noise rate is $\tau$. All experiments on those datasets with synthetic instance-dependent label noise are repeated five times. The real-world dataset Clothing1M has $1M$ images with real-world noisy labels and $10k$ images with clean labels for testing. In the experiments, we leave out $10\%$ of the noisy training samples as a noisy validation set for model selection.

**Baselines.** We compare the proposed IF method with (i) CE, which trains a standard deep network with the cross-entropy loss on noisy datasets; (ii) DMI, a novel information-theoretic robust loss function for instance-independent label noise (Xu et al., 2019); (iii) Forward (Patrini et al., 2017), Reweight (Liu & Tao, 2015), and T-Revision (Xia et al., 2019); (iv) part-dependent transition matrix (PTD) (Xia et al., 2020); (v) instance-level forward correction (ILFC) (Berthon et al., 2021). All these approaches utilize the NTM to correct the loss function.

**Implementations.** We choose standard neural networks and optimizers. Specifically, we use a ResNet-18 network for F-MNIST, a ResNet-34 network for SVHN and CIFAR-10. For the optimization, we first use SGD with $0.9$ momentum, $10^{-4}$ weight decay, $128$ batch size, $50$ epochs and an initial learning rate of $10^{-2}$ to initialize the network. Then, we adopt Adam optimizer and $5 \times 10^{-7}$ learning rate to learn the NTM following PTD (Xia et al., 2020). Once the NTM is obtained, we retrain the neural network. In the forward propagation, the Kalman transition matrix is obtained and then adopted to correct the loss function. In addition, the revision trick (Xia et al., 2019) is also adopted for our proposed IF method, which introduces the slack variables $\Delta W$ for the estimated

---

[2]The ideal estimated transition matrix is to approximate the empirical (actual) transition matrix $W^*(x)$, instead of the expectation of label transition matrix. Therefore, the estimation error is defined to characterize the bias–variance tradeoff for transition matrix estimation.

Table 1: Means and standard deviations (percentage) of classification accuracy with different instance-dependent label noise levels. Methods with "-F" adopt the Forward correction loss; methods with "-V" mean that the transition matrices are revised via the slack variable trick.

| Dataset | Method | IDN-10% | IDN-20% | IDN-30% | IDN-40% | IDN-50%. |
|---|---|---|---|---|---|---|
| SVHN | CE | $90.77 \pm 0.45$ | $90.23 \pm 0.62$ | $86.33 \pm 1.34$ | $65.66 \pm 1.65$ | $48.01 \pm 4.59$ |
| | DMI | $93.51 \pm 1.09$ | $93.22 \pm 0.62$ | $91.78 \pm 1.54$ | $69.34 \pm 2.45$ | $48.93 \pm 2.34$ |
| | Forward | $90.89 \pm 0.60$ | $90.65 \pm 0.27$ | $87.32 \pm 0.59$ | $78.46 \pm 2.58$ | $46.27 \pm 3.90$ |
| | Reweight | $92.49 \pm 0.44$ | $91.09 \pm 0.34$ | $90.25 \pm 0.77$ | $84.48 \pm 0.86$ | $45.46 \pm 3.56$ |
| | T-Revision | $94.24 \pm 0.53$ | $94.00 \pm 0.88$ | $93.01 \pm 0.83$ | $88.63 \pm 1.37$ | $49.02 \pm 4.33$ |
| | ILFC | $92.08 \pm 0.12$ | $91.67 \pm 0.16$ | $90.80 \pm 0.15$ | $89.16 \pm 0.67$ | $65.69 \pm 6.54$ |
| | PTD-F | $91.92 \pm 0.87$ | $90.21 \pm 1.01$ | $87.11 \pm 1.43$ | $81.10 \pm 2.78$ | $67.81 \pm 5.41$ |
| | PTD-F-V | $92.64 \pm 0.81$ | $93.94 \pm 0.25$ | $92.71 \pm 0.44$ | $90.71 \pm 0.91$ | $80.35 \pm 5.46$ |
| | PTM-F | $93.78 \pm 0.12$ | $92.55 \pm 0.40$ | $89.25 \pm 0.09$ | $81.89 \pm 2.82$ | $67.30 \pm 3.07$ |
| | PTM-F-V | $93.89 \pm 0.10$ | $92.72 \pm 0.18$ | $88.97 \pm 0.26$ | $81.14 \pm 2.06$ | $68.95 \pm 3.02$ |
| | IF-F | $\mathbf{94.70 \pm 0.14}$ | $\mathbf{94.01 \pm 0.31}$ | $\mathbf{93.33 \pm 0.15}$ | $91.60 \pm 0.29$ | $85.57 \pm 3.17$ |
| | IF-F-V | $94.63 \pm 0.12$ | $93.92 \pm 0.26$ | $93.21 \pm 0.20$ | $\mathbf{91.66 \pm 0.23}$ | $\mathbf{86.24 \pm 2.61}$ |
| CIFAR10 | CE | $74.49 \pm 0.29$ | $68.21 \pm 0.72$ | $60.48 \pm 0.62$ | $49.84 \pm 1.27$ | $38.86 \pm 2.71$ |
| | DMI | $75.02 \pm 0.45$ | $69.89 \pm 0.33$ | $61.88 \pm 0.64$ | $51.23 \pm 1.18$ | $41.45 \pm 1.97$ |
| | Forward | $73.45 \pm 0.23$ | $68.99 \pm 0.62$ | $60.21 \pm 0.75$ | $47.17 \pm 2.96$ | $40.75 \pm 2.09$ |
| | Reweight | $74.55 \pm 0.23$ | $68.42 \pm 0.75$ | $62.58 \pm 0.46$ | $50.12 \pm 0.96$ | $41.08 \pm 2.45$ |
| | T-Revision | $74.61 \pm 0.39$ | $69.32 \pm 0.64$ | $64.09 \pm 0.37$ | $50.38 \pm 0.87$ | $42.57 \pm 3.27$ |
| | ILFC | $80.22 \pm 0.33$ | $74.46 \pm 0.09$ | $73.27 \pm 0.24$ | $57.00 \pm 4.07$ | $36.27 \pm 0.69$ |
| | PTD-F | $79.77 \pm 0.91$ | $74.96 \pm 0.71$ | $70.68 \pm 0.81$ | $61.92 \pm 1.59$ | $45.34 \pm 4.67$ |
| | PTD-F-V | $80.08 \pm 0.86$ | $74.67 \pm 0.36$ | $71.66 \pm 1.05$ | $62.45 \pm 1.73$ | $46.16 \pm 4.48$ |
| | PTM-F | $78.16 \pm 0.36$ | $74.81 \pm 0.81$ | $70.03 \pm 0.38$ | $63.48 \pm 0.38$ | $51.03 \pm 3.08$ |
| | PTM-F-V | $78.58 \pm 0.31$ | $75.06 \pm 0.56$ | $69.83 \pm 0.58$ | $62.69 \pm 0.69$ | $50.53 \pm 3.41$ |
| | IF-F | $80.92 \pm 0.28$ | $\mathbf{79.58 \pm 0.52}$ | $74.34 \pm 0.86$ | $\mathbf{68.21 \pm 2.21}$ | $50.07 \pm 3.95$ |
| | IF-F-V | $\mathbf{80.94 \pm 0.43}$ | $79.54 \pm 0.45$ | $\mathbf{74.67 \pm 0.92}$ | $68.03 \pm 2.90$ | $\mathbf{52.34 \pm 1.31}$ |
| F-MNIST | CE | $88.54 \pm 0.31$ | $88.38 \pm 0.42$ | $84.22 \pm 0.35$ | $68.86 \pm 0.78$ | $51.42 \pm 0.66$ |
| | DMI | $91.98 \pm 0.62$ | $90.33 \pm 0.21$ | $84.81 \pm 0.44$ | $69.01 \pm 1.87$ | $51.64 \pm 1.78$ |
| | Forward | $89.05 \pm 0.43$ | $88.61 \pm 0.43$ | $84.27 \pm 0.46$ | $70.25 \pm 1.28$ | $57.33 \pm 3.75$ |
| | Reweight | $90.33 \pm 0.27$ | $89.70 \pm 0.35$ | $87.04 \pm 0.35$ | $80.29 \pm 0.89$ | $65.27 \pm 1.33$ |
| | T-Revision | $91.56 \pm 0.31$ | $90.68 \pm 0.66$ | $89.46 \pm 0.45$ | $87.21 \pm 1.20$ | $74.22 \pm 0.81$ |
| | ILFC | $91.84 \pm 0.04$ | $90.47 \pm 0.29$ | $87.07 \pm 2.89$ | $86.68 \pm 0.36$ | $65.93 \pm 0.29$ |
| | PTD-F | $91.13 \pm 0.34$ | $89.51 \pm 0.65$ | $89.01 \pm 0.40$ | $89.18 \pm 0.20$ | $75.37 \pm 3.63$ |
| | PTD-F-V | $91.71 \pm 0.28$ | $91.18 \pm 0.30$ | $90.62 \pm 0.08$ | $89.38 \pm 0.69$ | $77.80 \pm 7.29$ |
| | PTM-F | $91.39 \pm 0.45$ | $90.66 \pm 0.07$ | $88.48 \pm 0.09$ | $83.20 \pm 1.02$ | $63.85 \pm 3.92$ |
| | PTM-F-V | $91.73 \pm 0.32$ | $91.07 \pm 0.14$ | $88.87 \pm 0.62$ | $82.20 \pm 0.92$ | $64.23 \pm 4.73$ |
| | IF-F | $91.58 \pm 0.16$ | $91.23 \pm 0.09$ | $90.10 \pm 0.74$ | $89.37 \pm 0.26$ | $82.78 \pm 0.21$ |
| | IF-F-V | $\mathbf{92.08 \pm 0.06}$ | $\mathbf{91.73 \pm 0.19}$ | $\mathbf{90.82 \pm 0.33}$ | $\mathbf{90.32 \pm 0.12}$ | $\mathbf{83.33 \pm 0.73}$ |

Kalman transition matrix. Specifically, the slack variable $\Delta W$ is initialized as all zero elements in the experiments and can be optimized during the training. To guarantee valid transition matrices, We first project their negative entries of $W_{km}(\boldsymbol{x}) + \Delta W$ to zero and then normalize the summation of each row to be 1. Following (Xia et al., 2020), we do not use any data augmentation technique in the experiments. For Clothing1M, we use a ResNet-50 pre-trained model on ImageNet. We do not use any clean data to learn the transition matrices and classifiers. After the NTM $\hat{T}(\boldsymbol{x})$ is obtained according to (Xia et al., 2020), we use SGD with $0.9$ momentum, $10^{-3}$ weight decay, 32 batch size, and run with $10^{-3}$ learning rate and 10 epochs. For the evaluation, we adopt the test accuracy of the epoch with best accuracy on validation datasets.

## 4.2 EXPERIMENTAL RESULTS

**Accuracy on synthetic and real-world noisy datasets.** To answer **Q1**, Figure 3 shows the estimation errors of PTD, PTM (i.e., only adopt PTM in loss correction) and IF methods with noise rate IDN-10% to IDN-40%, where the estimation error is defined as the average difference of the transition probability $err = \frac{1}{N} \sum_{n=1}^{N} (1 - \hat{W}_{y_n \hat{y}_n})$. It can be observed that IF can achieve the lowest estimation error over all noise ratios[3]. Table 1 reports the classification accuracy on the SVHN, CIFAR-10 and F-MNIST with noise rate IDN-10% to IDN-50%, where the highest accuracy is bold faced. We can observe that IF achieves better performance than all baselines across all the three datasets and five noise rates. For example, in CIFAR-10, the improvements over the strongest baseline

---

[3]The transition matrix estimation for instance-dependent label noise is more challenging than the counterpart for class-dependent label noise (Li et al., 2021).

Table 2: Classification accuracy on *Clothing1M*. In the experiments, only noisy samples are exploited to train and validate the deep model.

| CE | Forward | Reweight | T-Revision | PTD-F | PTD-F-V | ELR | IF-F | IF-F-V |
|-------|---------|----------|------------|-------|---------|-------|-------|--------|
| 68.94 | 69.98 | 70.82 | 71.27 | 70.37 | 70.62 | 71.86 | 72.04 | **72.29** |

is $1.07\%$, $0.83\%$, $4.03\%$, $9.22\%$, $9.89\%$ for noise rate IDN-$10\%$ to IDN-$50\%$, respectively. The higher accuracy implies that the posterior information is more important for higher noise rates. In addition, the methods IF-F and IF-F-V are comparable across the three datasets and noise rates, which demonstrates that the revision on the transition matrix is not significantly beneficial compared with the method PTD. This observation validates the effectiveness of posterior information on transition matrix modification. For the real-world dataset Clothing1M, IF outperforms the baselines as shown in Table 2. Our IF method achieves $0.5\%$ improvements over the strongest baseline early-learning regularization (ELR) (Liu et al., 2020).

**Transition matrix estimation evaluation.** We study the transition matrix estimation error in the training data for IF and PTM to answer **Q2**. Figure 4 shows the training accuracy and estimation error of each epoch on the CIFAR10 dataset with noise rate IDN-$10\%$ to IDN-$40\%$, respectively. The shadow area represents the standard deviation. For all IDN rates, IF can achieve higher training accuracy and lower estimation error compared with PTM.

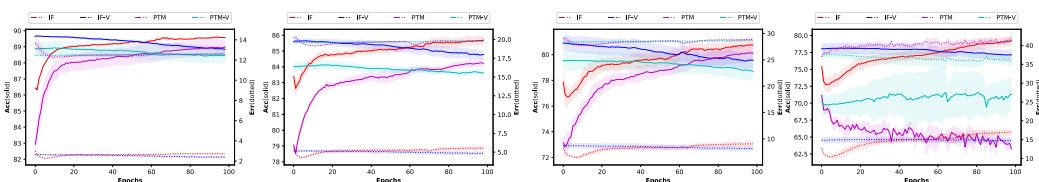

Figure 4: The training accuracy and transition matrix estimation error on the CIFAR-10 dataset with IDN noise $0.1$, $0.2$, $0.3$, and $0.4$.

**Stability on synthetic noisy datasets.** We study the training stability via inspecting the test accuracy over epochs to further answer **Q2**. Figure 5 shows the test accuracy with respect to epochs on the CIFAR10 dataset with noise rate IDN-$10\%$ to IDN-$40\%$, respectively. The shadow area represents the standard deviation of the accuracy on the same epoch across five experiments. We can clearly observe that, on both low-level and high-level noise, IF shows higher accuracy in a more stable way since the posterior information of each instance provides reliable guidance on loss correction.

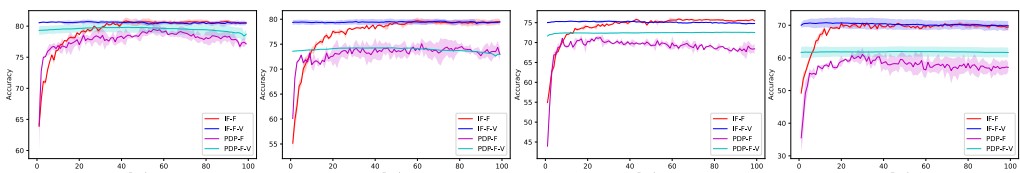

Figure 5: The test accuracy on the CIFAR-10 dataset with IDN noise $0.1$, $0.2$, $0.3$, and $0.4$.

## 5 CONCLUSIONS

This paper presents a novel notion of PTM and a simple yet effective learning paradigm named information fusion, which trains deep neural networks under instance-dependent label noise. The main goal of PTM is to characterize the label noise via limited observed noisy labels, which can bridge the empirical noisy and posterior empirical clean distributions. To effectively combine the benefits of NTM and PTM, we propose an information fusion approach to integrate both transition matrices to correct the loss function. Experiments on synthetic and real-world noisy datasets show that IF can achieve higher accuracy and more stable training compared with state-of-the-art methods, especially for high noise rates. In the future, we can extend to incorporate more prior knowledge of the transition matrix, e.g., sparsity or low rank, into the end-to-end learning algorithm.

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

In this Appendix, we provide A) theoretical analysis, B) Reproducibility, C) Additional Experimental Results, and D) Related Works.

# A  THEORETICAL ANALYSIS

## A.1  PROOF OF THEOREM 3.1

Based on the definition of posterior reweight loss, we have $l_{p-rew}\big(\tilde{y}, f(\boldsymbol{x})\big) = \sum_{y=1}^{c} \mathbb{P}(Y = y|\tilde{Y} = \tilde{y}, X = \boldsymbol{x})l(y, f(\boldsymbol{x}))$. For noisy underlying distribution, the expected risk can be calculated as follows,

$$
\begin{aligned}
& \mathbb{E}_{(\boldsymbol{x}, \tilde{y}) \sim \mathbb{P}(X, \tilde{Y})}\Big[l_{p-rew}\big(\tilde{y}, f(\boldsymbol{x})\big)\Big] \\
= & \int_{\boldsymbol{x}} \sum_{\tilde{y}=1}^{c} \mathbb{P}(X = \boldsymbol{x}, \tilde{Y} = \tilde{y})l_{p-rew}\big(\tilde{y}, f(\boldsymbol{x})\big)\mathrm{d}\boldsymbol{x} \\
= & \int_{\boldsymbol{x}} \sum_{\tilde{y}=1}^{c} \mathbb{P}(X = \boldsymbol{x}, \tilde{Y} = \tilde{y}) \sum_{y=1}^{c} \mathbb{P}(Y = y|\tilde{Y} = \tilde{y}, X = \boldsymbol{x})l\big(y, f(\boldsymbol{x})\big)\mathrm{d}\boldsymbol{x} \\
= & \int_{\boldsymbol{x}} \sum_{\tilde{y}=1}^{c} \sum_{y=1}^{c} \mathbb{P}(X = \boldsymbol{x}, Y = y, \tilde{Y} = \tilde{y})l\big(y, f(\boldsymbol{x})\big)\mathrm{d}\boldsymbol{x} \\
= & \int_{\boldsymbol{x}} \sum_{y=1}^{c} \mathbb{P}(X = \boldsymbol{x}, Y = y)l\big(y, f(\boldsymbol{x})\big)\mathrm{d}\boldsymbol{x} = \mathbb{E}_{(\boldsymbol{x}, y) \sim \mathbb{P}(X, Y)}\Big[l\big(y, f(\boldsymbol{x})\big)\Big], \quad (11)
\end{aligned}
$$

which implies that posterior reweight loss over noisy underlying distribution equals to cross entropy loss over clean underlying distribution and thus posterior reweight loss induces to statistically consistent classifier.

For the anchor point samples $(\boldsymbol{x}_{ap}, \tilde{y}_{ap})$ with underlying clean probability $\mathbb{P}(Y = y_{ap}|X = \boldsymbol{x}_{ap})$, based on equation (2), the PTM value satisfies $W_{y_{ap}, \tilde{y}_{ap}}(\boldsymbol{x}) = 1$ and $W_{k, \tilde{y}_{ap}} = 0$ for any $k \neq y_{ap}$, which implies that empirical distribution mismatch on anchor points samples can be completely mitigated. As for empirical distribution, the empirical risk satisfies

$$
l_{p-rew}\big(\tilde{y}_{ap}, f(\boldsymbol{x}_{ap})\big) = l(y_{ap}, f(\boldsymbol{x}_{ap})). \quad (12)
$$

For noisy empirical distribution, the expected risk satisfies

$$
\begin{aligned}
\mathbb{E}_{(\boldsymbol{x}, \tilde{y}) \sim \hat{\mathbb{P}}_{\tilde{D}}(X, \tilde{Y})}\Big[l_{p-rew}\big(\tilde{y}, f(\boldsymbol{x})\big)\Big] & = \frac{1}{N} \sum_{n=1}^{N} l_{p-rew}\big(\tilde{y}_i, f(\boldsymbol{x}_i)\big) \\
& = \frac{1}{N} \sum_{n=1}^{N} \sum_{y=1}^{c} \mathbb{P}(Y = y|\tilde{Y} = \tilde{y}_n, X = \boldsymbol{x}_n)l(y, f(\boldsymbol{x})) \\
& = \mathbb{E}_{(\boldsymbol{x}, y) \sim \hat{\mathbb{P}}(X, Y|\tilde{D})}\Big[l(y_n, f(\boldsymbol{x}_n))\Big]. \quad (13)
\end{aligned}
$$

which implies that posterior reweight loss over noisy empiricial distribution equals to cross entropy loss over posterior clean empirical distribution.

## A.2  PROOF OF LEMMA 3.2

Note that the loss function $l(y, f)$ is convex with respect to $f$ and $\sum_{i=1}^{c} W_{i, \tilde{y}}(\boldsymbol{x}) = \sum_{i=1}^{c} \mathbb{P}(Y = i|\tilde{Y} = \tilde{y}, X = \boldsymbol{x}) = 1$ for any $\tilde{y}$ and $\boldsymbol{x}$, according to Jensen's inequality, we have

$$
\begin{aligned}
l_{p-rew}\big(\tilde{y}, f(\boldsymbol{x})\big) & = \sum_{y=1}^{c} W_{i, \tilde{y}}(\boldsymbol{x})l(y, f(\boldsymbol{x})) \\
& \geq l\Big(\tilde{y}, \sum_{i=1}^{c} W_{i, \tilde{y}}(\boldsymbol{x})f(\boldsymbol{x})\Big) = l_{p-fw}\big(\tilde{y}, f(\boldsymbol{x})\big), \quad (14)
\end{aligned}
$$

which shows the relation between posterior reweight loss and posterior forward loss.

### A.3 Proof of Theorem 3.3

Based on Lemma 3.2 and Theorem 3.1, we have the inequality on the expected risk as follows,

$$\mathbb{E}_{(\boldsymbol{x},\tilde{y})\sim\mathbb{P}(X,\tilde{Y})}\Big[l_{p-fw}\big(\tilde{y},f(\boldsymbol{x})\big)\Big] \leq \mathbb{E}_{(\boldsymbol{x},\tilde{y})\sim\mathbb{P}(X,\tilde{Y})}\Big[l_{p-rew}\big(\tilde{y},f(\boldsymbol{x})\big)\Big] = \mathbb{E}_{(\boldsymbol{x},y)\sim\mathbb{P}(X,Y)}[l(y,f(\boldsymbol{x}))],$$

Define the optimal classifier $f^*$ to minimize expected risk for underlying clean distribution and cross entropy loss as

$$f^* = \arg\min_f R_{\mathbb{P}(X,Y)}(f) = \arg\min_f \mathbb{E}_{(\boldsymbol{x},y)\sim\mathbb{P}(X,Y)}[l(y,f(\boldsymbol{x}))], \tag{15}$$

then $R_{\mathbb{P}(X,Y)}(f^*) = 0$ and for $l_{p-rew}\big(\tilde{y},f^*(\boldsymbol{x})\big) = 0$ for any $\boldsymbol{x}$. Note that the expected risk for posterior forward loss is non-negative and not larger than that for posterior reweight loss, we also have $l_{p-fw}\big(\tilde{y},f^*(\boldsymbol{x})\big) = 0$ for any $\boldsymbol{x}$. In addition, the difference of expected risk for two different classifiers satisfies

$$\mathbb{E}_{(\boldsymbol{x},\tilde{y})\sim\mathbb{P}(X,\tilde{Y})}\Big[l_{p-fw}\big(\tilde{y},f^*(\boldsymbol{x})\big)\Big] - \mathbb{E}_{(\boldsymbol{x},\tilde{y})\sim\mathbb{P}(X,\tilde{Y})}\Big[l_{p-fw}\big(\tilde{y},f(\boldsymbol{x})\big)\Big]$$

$$= -\mathbb{E}_{(\boldsymbol{x},\tilde{y})\sim\mathbb{P}(X,\tilde{Y})}\Big[l_{p-fw}\big(\tilde{y},f(\boldsymbol{x})\big)\Big] \leq 0, \tag{16}$$

which implies that the classifier $f^*$ can also minimize expected risk for underlying noisy distribution and posterior reweight loss. This completes the proof.

### A.4 Proof of Theorem 3.4

Aiming to standardize the constraint, we vectorize $W(\boldsymbol{x})$ as $c^2 \times 1$ dimension $\mathbf{w}_{vec}(\boldsymbol{x}) := [W_{11}(\boldsymbol{x}),\cdots,W_{c1}(\boldsymbol{x}),\cdots,W_{1c}(\boldsymbol{x}),\cdots,W_{cc}(\boldsymbol{x})]^\top$ and define the coefficient matrix $A(\boldsymbol{x})$ as follows,

$$A(\boldsymbol{x}) = \begin{bmatrix} \hat{\mathbb{P}}^\top(\tilde{\mathbf{Y}}|\boldsymbol{x}) & \cdots & \mathbf{0} & \cdots & \mathbf{0} \\ \mathbf{0} & \hat{\mathbb{P}}^\top(\tilde{\mathbf{Y}}|\boldsymbol{x}) & \mathbf{0} & \cdots & \mathbf{0} \\ \vdots & \vdots & \ddots & \vdots & \vdots \\ \mathbf{0} & \mathbf{0} & \mathbf{0} & \cdots & \hat{\mathbb{P}}^\top(\tilde{\mathbf{Y}}|\boldsymbol{x}) \end{bmatrix}_{c\times c^2} \tag{17}$$

where $\mathbb{P}^\top(\tilde{\mathbf{Y}}|\boldsymbol{x}) = [\mathbb{P}(\tilde{Y}=1|\boldsymbol{x}),\cdots,\mathbb{P}(\tilde{Y}=c|\boldsymbol{x})]_{1\times c}$. Therefore, the constraint $W^\top(\boldsymbol{x})\hat{\mathbb{P}}(\tilde{Y}|\boldsymbol{x}) = f(\boldsymbol{x})$ can be transformed as $A(\boldsymbol{x})\mathbf{w}_{vec}(\boldsymbol{x}) = f(\boldsymbol{x})$, which is a least-norm problem with undetermined equations constraint. Note that matrix $A(\boldsymbol{x})$ is full column rank, we can prove that the least-norm solutions is

$$\mathbf{w}_{vec}^*(\boldsymbol{x}) = A^\top(\boldsymbol{x})\Big(A(\boldsymbol{x})A^\top(\boldsymbol{x})\Big)^{-1}f(\boldsymbol{x}). \tag{18}$$

Considering any possible solution $\mathbf{w}_{vec}(\boldsymbol{x})$ satisfying the constraint $A(\boldsymbol{x})\mathbf{w}_{vec}(\boldsymbol{x}) = f(\boldsymbol{x})$, then $A(\boldsymbol{x})[\mathbf{w}_{vec}(\boldsymbol{x}) - \mathbf{w}_{vec}^*(\boldsymbol{x})] = \mathbf{0}$ and

$$[\mathbf{w}_{vec}(\boldsymbol{x}) - \mathbf{w}_{vec}^*(\boldsymbol{x})]^\top \mathbf{w}_{vec}^*(\boldsymbol{x}) = [\mathbf{w}_{vec}(\boldsymbol{x}) - \mathbf{w}_{vec}^*(\boldsymbol{x})]^\top A^\top(\boldsymbol{x})\Big(A(\boldsymbol{x})A^\top(\boldsymbol{x})\Big)^{-1}f(\boldsymbol{x})$$

$$= \Big[A(\boldsymbol{x})[\mathbf{w}_{vec}(\boldsymbol{x}) - \mathbf{w}_{vec}^*(\boldsymbol{x})]\Big]^\top \Big(A(\boldsymbol{x})A^\top(\boldsymbol{x})\Big)^{-1}f(\boldsymbol{x}) = \mathbf{0},$$

which implies that the vectors $\mathbf{w}_{vec}(\boldsymbol{x}) - \mathbf{w}_{vec}^*(\boldsymbol{x})$ and $\mathbf{w}_{vec}^*(\boldsymbol{x})$ are perpendicular. Thus, the $L_2$ norm of $\mathbf{w}_{vec}(\boldsymbol{x})$ satisfies

$$||\mathbf{w}_{vec}(\boldsymbol{x})||^2 = ||\mathbf{w}_{vec}^*(\boldsymbol{x}) + \mathbf{w}_{vec}(\boldsymbol{x}) - \mathbf{w}_{vec}^*(\boldsymbol{x})||^2$$

$$= ||\mathbf{w}_{vec}^*(\boldsymbol{x})||^2 + ||\mathbf{w}_{vec}(\boldsymbol{x}) - \mathbf{w}_{vec}^*(\boldsymbol{x})||^2 \geq ||\mathbf{w}_{vec}^*(\boldsymbol{x})||^2. \tag{19}$$

which shows that the equation (18) is the optimal solution. Via direct calculation, it is easy to obtain that $A(\boldsymbol{x})A^\top(\boldsymbol{x}) = \sum_{j=k}^c \hat{\mathbb{P}}^2(\tilde{Y}=k|\boldsymbol{x})\mathbf{I}_{c\times c}$. Therefore, the element of PTM satisfies

$$W_{ij}^*(\boldsymbol{x}) = \sum_{k=1}^c \hat{\mathbb{P}}^2(\tilde{Y}=k|\boldsymbol{x})\hat{\mathbb{P}}(\tilde{Y}=i|\boldsymbol{x})f_j(\boldsymbol{x}), \tag{20}$$

i.e., $W^*(\boldsymbol{x}) = \frac{f(\boldsymbol{x})\hat{\mathbb{P}}(\tilde{Y}|\boldsymbol{x})^\top}{||\hat{\mathbb{P}}(\tilde{Y}|\boldsymbol{x})||_2^2}$ and completes the proof.

A.5    PROOF OF THEOREM 3.5

Considering the general reconstruction error relationship, the trace of the covariance matrix for reconstruction error satisfies $Tr(e_{\hat{T}}(\boldsymbol{x})e_{\hat{W}}^{\top}(\boldsymbol{x})) = cov(x)\sqrt{\sigma_T(\boldsymbol{x})\sigma_W(\boldsymbol{x})}$. For Kalman transition matrix, the reconstruction deviation satisfies $W^*(\boldsymbol{x}) - W_{km}(\boldsymbol{x}) = (1-\lambda)e_T(\boldsymbol{x}) + \lambda e_W(\boldsymbol{x})$. For simplicity, we remove the instance $\boldsymbol{x}$ to abbreviate the notations. Therefore, the reconstruction error is given by

$$
\begin{aligned}
\mathcal{L}_{mse}(\lambda) &= \mathbb{E}\Big[Tr\Big(\big((1-\lambda)e_T + \lambda e_W\big)\big((1-\lambda)e_T + \lambda e_W\big)^{\top}\Big)\Big] \\
&= (1-\lambda)^2\sigma_T + \lambda^2\sigma_W + 2\lambda(1-\lambda)cov\sqrt{\sigma_T\sigma_W},
\end{aligned}
\tag{21}
$$

For such quadratic function, it is easy to obtain the optimal parameter via the local minimal

$$
\frac{\partial\mathcal{L}_{mse}}{\partial\lambda} = 2(1-\lambda)\sigma_T + 2\lambda\sigma_W + 2(1-2\lambda)cov\sqrt{\sigma_T\sigma_W} = 0,
\tag{22}
$$

and thus the optimal weight is given by $\lambda^* = \frac{\sigma_T - cov\sqrt{\sigma_T\sigma_W}}{\sigma_T + \sigma_W - 2cov\sqrt{\sigma_T\sigma_W}}$. Plugging in the optimal weight into the reconstruction error, we obtain the minimal reconstruction error is given by

$$
\begin{aligned}
\mathcal{L}_{mse}(\lambda^*) &= \frac{1}{[\sigma_T + \sigma_W - 2cov\sqrt{\sigma_T\sigma_W}]^2}\Big\{\sigma_T\big[\sigma_W - cov\sqrt{\sigma_T\sigma_W}\big]^2 \\
&\quad + \sigma_W\big[\sigma_T - cov\sqrt{\sigma_T\sigma_W}\big]^2 \\
&\quad + 2\big[\sigma_W - cov\sqrt{\sigma_T\sigma_W}\big]\big[\sigma_T - cov\sqrt{\sigma_T\sigma_W}\big]cov\sqrt{\sigma_T\sigma_W}\Big\}
\end{aligned}
\tag{23}
$$

Aiming to investigate the relation between minimal reconstruction error for IF, PTM and PTD, we adopt variable substitution $t = \sigma_T + \sigma_W - 2cov\sqrt{\sigma_T\sigma_W} \geq 0$ to further simplify minimal reconstruction error $\mathcal{L}_{mse}(\lambda^*)$, then we have

$$
\begin{aligned}
\mathcal{L}_{mse}(\lambda^*) &= \frac{\sigma_T[(\sigma_W - \sigma_T) + t]}{4t^2} + \frac{\sigma_W[(\sigma_T - \sigma_W) + t]}{4t^2} \\
&\quad + \frac{[(\sigma_T - \sigma_W) + t][(\sigma_W - \sigma_T) + t][(\sigma_W + \sigma_T) - t]}{4t^2} \\
&= \frac{\sigma_W + \sigma_T}{2} - \big[\frac{t}{4} + \frac{(\sigma_W - \sigma_T)^2}{4t}\big] \\
&\leq \frac{\sigma_W + \sigma_T}{2} - 2\sqrt{\frac{t}{4}\cdot\frac{(\sigma_W - \sigma_T)^2}{4t}} = \min\{\sigma_W, \sigma_T\}
\end{aligned}
\tag{24}
$$

where the inequality holds according to arithmetic mean-geometric mean inequality. Therefore, the linear combination can provably achieve lower estimation error for general $e_T(x)$ and $e_W(x)$, either correlated or independent.

# B    REPRODUCIBILITY

## B.1    DATASET STATISTICS

For fairly comparison with previous work, we performance the image classification task on the three manually corrupted datasets, i.e., F-MNIST, SVHN, CIFAR-10, and one real-world noisy dataset Clothing1M. They have been widely adopted to study label noise problem. The detailed statistics are listed in Table 3.

Table 3: Dataset statistics on F-MNIST, SVHN, CIFAR-10, and Clothing1M

|  | F-MNIST | SVHN | CIFAR-10 | Clothing1M |
| --- | --- | --- | --- | --- |
| # training/noisy images | 60,000 | 73,257 | 50,000 | 1,000,000 |
| # test/clean images | 10,000 | 26,032 | 10,000 | 10,000 |
| label noise | synthetic | synthetic | synthetic | real-world |

## B.2 RUNNING ENVIRONMENT

All baselines and IF approaches are implemented in PyTorch, and tested on a machine with AMD EPYC 7282 16-core processors, 4 GeForce GTX-3090 Ti GPUs with 24GB memory size.

## B.3 ALGORITHMS

We summarize the IF algorithm to correct the loss and provide the pseudo codes in Algorithm 1. For instance-dependent Label noise generation, we provide pseudo codes with controllable noise rate are provided in Algorithm 2. This algorithm follows the state-of-the-art method (Xia et al., 2020; Zhu et al., 2021a), where the overall noise rate is $\tau$.

---

**Algorithm 1:** Information Fusion Algorithm

**Input** : Noisy dataset $\tilde{D} = (\boldsymbol{x}_n, \tilde{y}_n)_{n=1}^N$; Noisy validation data; tolerant epochs $t$.
**Output :** The robust neural network over noisy label.

1   Estimate the NTM $\hat{T}(\boldsymbol{x})$ according to (Xia et al., 2020) ;
2   **while** *Validation accuracy has increases in the last $t$ epochs* **do**
3      Calculate the PTM $\hat{W}(\boldsymbol{x})$ based on Equation (8) ;
4      Obtain the uncertainty for noise and PTM ;
5      Calculate the Kalman gain and Kalman transition matrix based on Equations (9) and (10) ;
6      Correct the loss function as posterior forward loss with Kalman transition matrix $W_{km}(\boldsymbol{x})$ ;
7   **end**

---

**Algorithm 2:** Instance-Dependent Label Noise Generation

**Input** : Clean samples $(\boldsymbol{x}_n, y_n)_{n=1}^N$; Overall noise rate $\tau$; Number of classes $c$; Size of input features: $1 \times d$.
**Output :** Noisy samples $(\boldsymbol{x}_i, \tilde{y}_n)_{n=1}^N$.

1   Sample flip rate $q_i$ based on truncated normal distribution $\mathcal{N}(\tau, 0.1^2, [0, 1])$ ;
2   Sample the projection matrix $W \in \mathcal{R}^{d \times K}$ based on normal distribution $\mathcal{N}(0, 1^2)$;
3   **for** $n = 1$ *to* $N$ **do**
4      $p = \boldsymbol{x}_n W$ // Genrate instance dependent flip rate ;
5      $p_{y_n} = -\infty$ // Control the diagonal elements of the transition matrix ;
6      $p = q_n \cdot \text{softmax}(p)$ // Make the sum of the off-diagonal elements to be $q_n$ ;
7      $p_{y_n} = 1 - q_n$ // Set the diagonal element to be $1 - q_n$ ;
8   **end**

---

## B.4 LEARNING NOISE TRANSITION MATRIX VIA DECOMPOSITION

For the instance-dependent label noise, the NTM may be different for different samples. Thus, the complexity of NTM estimation is $O(Nc^2)$, which leads to ill-posed estimation problem. (Xia et al., 2020) introduces a part-dependent decomposition on the NTM, i.e., the noise of an instance depends only on its parts. Specifically, given $L$ part-dependent transition matrix, e.g, $P^l \in [0, 1]^{c \times c}$, where $j = 1, \cdots, L$, the instance-dependent matrix can be approximated by a combination of part-dependent transition matrix and the combination coefficients, defined as $\boldsymbol{h}(\boldsymbol{x}) \in [0, 1]^L$, depend on the feature vectors. Then the instance-dependent transition matrix $W(\boldsymbol{x})$ can be approximated as follows

$$\hat{T}(\boldsymbol{x}) = \sum_{l=1}^L \boldsymbol{h}_j(\boldsymbol{x}) P^l. \tag{25}$$

where $\boldsymbol{h}_j(\boldsymbol{x})$ represents the $j$-th elements of $\boldsymbol{h}(\boldsymbol{x})$ and satisfies the normalization condition $||\boldsymbol{h}(\boldsymbol{x})|| = 1$ for any features vector to maintain that the summation of column is 1 for NTM.

In this work, we follow (Xia et al., 2020) to learn the NTM in three steps: (i) learning combination coefficients $\boldsymbol{h}(\boldsymbol{x})$ via parts-based features representation (Lee & Seung, 1999; Tao et al., 2017); (ii)

learning the instance-dependent transition matrix by exploiting anchor points (Liu & Tao, 2015; Xia et al., 2019); (iii) learning the part-dependent transition matrices via minimization of reconstruction error for instance-dependent transition matrix.

(Xia et al., 2020) justifies the part-dependent decomposition assumption via parts-based representations. For example, (Hosseini-Asl et al., 2016) shows that the input features rely on parts-based representations in object recognition. Thus, it is natural to approximate the label noise in the part level. In display advertising, the delay feedback of the label (Ktena et al., 2019; Yasui et al., 2020) introduces label noise. Earlier the data is generated, the less probability the feedback exists. Thus, the label noise highly depends on part of features.

Since the instance-dependent NTM may be poorly learned, the slack variable trick (Xia et al., 2019; 2020) is also adopted to modify the instance-dependent transition matrix, which significantly improve the accuracy. However, the slack variable trick ignores the posterior information for each instance, which limits the accuracy. In addition, the variance of accuracy in (Xia et al., 2020) is large since there is not any guide information to modify the instance-dependent NTM except minimization of the corrected loss. In this paper, we claim that the measurement information is useful to modify the NTM. We will introduce how to estimate the PTM and information fusion in the next two subsections.

### B.5 Implementation Details

**Warm-up Training Strategy**    Aiming to estimate the transition matrix, we adopt warm-up training strategy that training the neural network via cross entropy loss with noisy data. we use SGD as the optimizer with 0.9 momentum, 128 batch size, and 10 epochs. With the benefit of memorization effect (Arpit et al., 2017), warm-up training with appropriate epochs can guarantee that the predicted probability of neural network is reliable and can approximate underlying clean probability.

**Training with Corrected Loss**    Our proposed PTM estimation relies on the neural network predicted probability. Although the warm-up training can guarantee the prediction accuracy for clean instances with the benefits of memorization effect, the prediction of the PTM estimation may still be not reliable. To ease the dilemma, we adopt sample selection to choose the reliable prediction and thus guarantee the reliable PTM estimation for specific instance. Similar to (Cheng et al., 2021a), the small-loss criteria is adopted to select the instances with small training loss for training. Specifically, we select the instance via the weighted loss if the instance loss is less than the average loss over all classes.

For the instance satisfying the small loss criterion, we have confidence that the instance is clean, which means the PTM is identity. To put it simply, we sum the estimated PTM and identity and then adopt column normalization. It is worth noting that the noisy labels are adopted in forward propagation, which is highly different with previous transition matrix-based works (Xia et al., 2019; Liu & Tao, 2015; Patrini et al., 2017; Xia et al., 2020). Compared with hard sample selection based methods (Cheng et al., 2021a), our method achieves the soft sample selection and adaptively correct the loss for "clean" instance.

## C    Additional Experimental Results

### C.1    Stability on Synthetic Noisy Datasets

We study the training stability via inspecting the test accuracy over epochs. Figures 6 and 7 show the test accuracy with respect to epochs on SVHN and F-MNIST datasets with noise rate IDN-$10\%$ to IDN-$40\%$. It is seen that our proposed method IF can achieve better accuracy and stable training compare with baseline PTD, which implies that the PTM fusion can decrease the transition matrix estimation error and improve the training stability. T-Revision trick (-V method) proposed in (Xia et al., 2019) aims to compensate the NTM and thus improve the accuracy performance. It is seen that PTD-F-V can achieve better performance than PTD-F, which verifies that the learned NTM is not very accurate. For our proposed method, IF-V can achieve comparable performance than IF-F and demonstrates that transition matrix revision trick can not improve the performance since PTM has already been employed to compensate the transition matrix error.

Compared with different noise rates, it is seen that the accuracy of PTD method may decrease with respect to epochs, especially for large noise rate. Under large noise rate, the number of clean samples

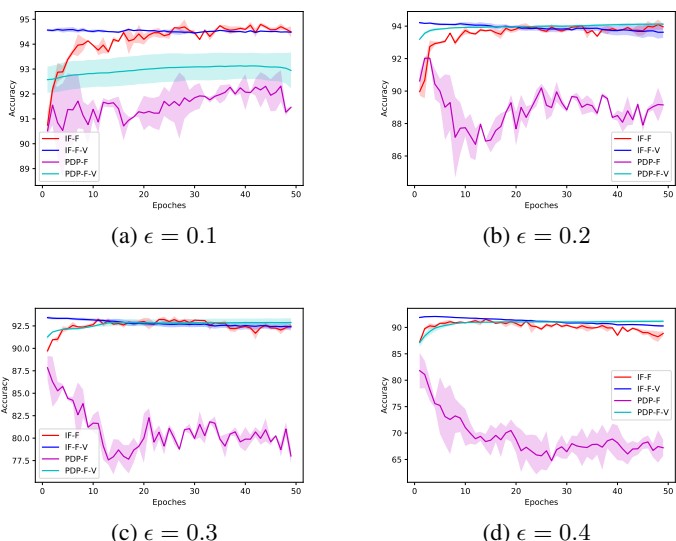

Figure 6: The test accuracy on SVHN datasets with different levels of IDN noise.

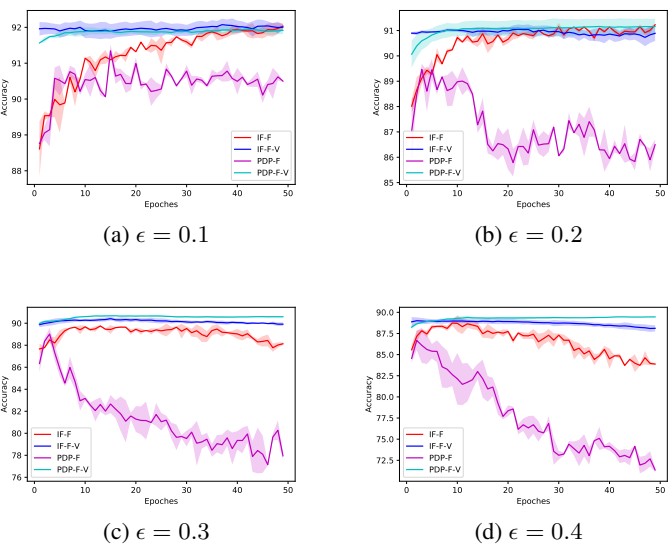

Figure 7: The test accuracy on F-MNIST datasets with different levels of IDN noise.

is limited and the neural network may already overfit noisy samples during warm-up training and thus hurt the transition matrix estimation. In this scenario, PTD method does not have the capability to adjust such poor NTM estimation and thus leads to the accuracy drop during training. Our proposed method IF and T-Revision trick both adjust the estimated transition matrix and mitigate the accuracy drop.

## C.2 EXPERIMENTAL RESULTS ON HIGH NOISE RATIO

We investigate the performance of IF for a high noise ratio compared with PTM and PTD. Table. 4 shows the test accuracy for high noise ratio IDN-70%, IDN-80% and IDN-90% in CIFAR 10 dataset. We have two observations: (1) Comparing with PTD at the noise ratios of IDN-80% and IDN-90%,

our PTM could still outperform PTD. Although the network output is not perfect, the performance deterioration of PTD is still higher than PTM. (2) Our IF consistently delivers the most superior performance no matter under the low or high noise. That is because IF adaptively and linearly combines NTM and PTM in instance level and thus achieves more accurate estimation.

Table 4: Means and standard deviations (percentage) of classification accuracy with high instance-dependent label noise levels for CIFAR10 dataset. Methods with "-F" adopts the Forward correction loss; methods with "-V" means that the transition matrices are revised via slack variable trick.

| Method | IDN-70% | IDN-80% | IDN-90% |
|--------|---------|---------|---------|
| PTD-F | $20.25 \pm 1.56$ | $13.48 \pm 0.81$ | $9.22 \pm 0.23$ |
| PTD-F-V | $20.35 \pm 1.56$ | $13.58 \pm 0.80$ | $9.44 \pm 0.24$ |
| PTM-F | $16.79 \pm 3.86$ | $14.13 \pm 0.34$ | $10.34 \pm 0.71$ |
| PTM-F-V | $18.95 \pm 2.89$ | $13.89 \pm 0.42$ | $10.57 \pm 0.82$ |
| IF-F | $\mathbf{21.26 \pm 1.46}$ | $15.78 \pm 1.65$ | $10.58 \pm 0.53$ |
| IF-F-V | $21.09 \pm 0.45$ | $\mathbf{16.72 \pm 0.61}$ | $\mathbf{10.86 \pm 0.40}$ |

## C.3 Experimental Results on More Baselines

Although many transition matrix-based methods are compared in Table. 1, we are also interested in several more baselines, including label smoothing (LS) (Szegedy et al., 2016), early-learning regularization (ELR) (Liu et al., 2020), (Berthon et al., 2021), progressive label correction (PLC) (Zhang et al., 2021a), confidence regularized sample sieve (CORES) (Cheng et al., 2021a). It could be observed that our method can still achieve state-of-the-art performance compared with all other baselines. Table. 5 demonstrates the test accuracy over IDN-10% to IDN-50% label noise ratio in CIFAR 10. It could be observed that our method can still achieve state-of-the-art performance, based on which we can confidently validate the effectiveness of our method.

Table 5: Means and standard deviations (percentage) of classification accuracy with more baselines for CIFAR10 dataset. Methods with "-F" adopts the Forward correction loss; methods with "-V" means that the transition matrices are revised via slack variable trick.

| Method | IDN-10% | IDN-20% | IDN-30% | IDN-40% | IDN-50% |
|--------|---------|---------|---------|---------|---------|
| LS | $63.19 \pm 0.25$ | $57.67 \pm 0.10$ | $49.50 \pm 0.17$ | $41.57 \pm 1.85$ | $34.89 \pm 2.47$ |
| ELR | $66.20 \pm 0.09$ | $60.58 \pm 0.75$ | $51.65 \pm 0.33$ | $43.83 \pm 1.61$ | $34.44 \pm 0.12$ |
| PLC | $66.38 \pm 0.61$ | $60.42 \pm 0.65$ | $51.60 \pm 0.58$ | $42.77 \pm 0.50$ | $36.00 \pm 1.73$ |
| CORES | $67.39 \pm 0.44$ | $60.85 \pm 0.38$ | $51.47 \pm 0.80$ | $43.96 \pm 0.68$ | $34.85 \pm 1.66$ |
| PTD-F | $79.77 \pm 0.91$ | $74.96 \pm 0.71$ | $70.68 \pm 0.81$ | $61.92 \pm 1.59$ | $45.34 \pm 4.67$ |
| PTD-F-V | $80.08 \pm 0.86$ | $74.67 \pm 0.36$ | $71.66 \pm 1.05$ | $62.45 \pm 1.73$ | $46.16 \pm 4.48$ |
| PTM-F | $78.16 \pm 0.36$ | $74.81 \pm 0.81$ | $70.03 \pm 0.38$ | $63.48 \pm 0.38$ | $51.03 \pm 3.08$ |
| PTM-F-V | $78.58 \pm 0.31$ | $75.06 \pm 0.56$ | $69.83 \pm 0.58$ | $62.69 \pm 0.69$ | $50.53 \pm 3.41$ |
| IF-F | $80.92 \pm 0.28$ | $\mathbf{79.58 \pm 0.52}$ | $74.34 \pm 0.86$ | $\mathbf{68.21 \pm 2.21}$ | $50.07 \pm 3.95$ |
| IF-F-V | $\mathbf{80.94 \pm 0.43}$ | $79.54 \pm 0.45$ | $\mathbf{74.67 \pm 0.92}$ | $68.03 \pm 2.90$ | $\mathbf{52.34 \pm 1.31}$ |

# D Related work

**Label Noise Model** There are three types of label noise model, including the random classification noise (RCN) model (Manwani & Sastry, 2013; Natarajan et al., 2013), the class-conditional label noise (CCN) model (Patrini et al., 2017; Xia et al., 2019; Zhang & Sabuncu, 2018; Liu & Guo, 2020) and the e instance-dependent label noise (IDN) model (Cheng et al., 2020; Xia et al., 2020; Cheng et al., 2021a; Berthon et al., 2021; **?**). Specifically, RCN means that each label is flipped independently with a constant probability, CCN assumes that the flip probability (noise rates) are the same for the instance with the same clean labels. IDN is the most general case, where the flip probability depends on its instance.

**Loss Correction via Transition Matrix** The transition matrix bridges the gap between the model predicted probability for noisy and clean data, which can achieve *risk-consistent* classifier with label

noise. The transition matrix can be estimated via cross-validation method (Natarajan et al., 2013), anchor points assumption (Xia et al., 2019; Yu et al., 2018; Yao et al., 2020). Subsequently, the estimated transition matrix can be adopted to correct the loss function via forward correction (Patrini et al., 2017), backward correction (Patrini et al., 2017) and reweight (Xia et al., 2019; Liu & Tao, 2015).

**Loss Correction via Sample Selection**   Due to the experiments observation– the neural network learns clean instances first (Arpit et al., 2017) and memorizes the instances gradually, there are lots of methods attempting to identify mislabeled training examples and then filter them out (Brodley & Friedl, 1999; Jiang et al., 2018; Han et al., 2018b; Yu et al., 2019; Angelova et al., 2005; Huang et al., 2019; Li et al., 2019). The core idea is to select the "small loss" samples as clean one.

