# OpenReview forum: "An Information Fusion Approach to Learning with Instance-Dependent Label Noise"
_ICLR.cc/2022/Conference — ICLR 2022 Poster_

### Official Review · Reviewer_cR7o · 2021-10-17

**Correctness:** 4
**Technical Novelty And Significance:** 3
**Empirical Novelty And Significance:** 2
**Recommendation:** 8
**Confidence:** 5

**Details Of Ethics Concerns:**

I do not have ethics concerns on this paper.


**Main Review:**

Pros:
1. The argument in this paper, namely IDN is indeed caused by empirical distribution rather than underlying real distribution, is interesting.
2. The proposed estimation method for PTM and the fusion of PTM and NTM via Kalman filtering are new.
3. Theoretical analyses in this paper are sufficient.

Cons:
1. The title of this paper can be more concise.
2. More related baseline methods are suggested to be added.



**Summary Of The Paper:**

This paper proposes a new information fusion approach to deal with the instance-dependent label noise (IDN). Specifically, the authors claim that the essential problem caused by IDN is empirical, instead of underlying, data distribution mismatch during training, which I think is new and interesting. Therefore, the authors fuse posterior transition matrix (PTM) and noise transition matrix (NTM) and design an unbiased risk estimator.

**Summary Of The Review:**

Generally, I feel that the claim in this paper is reasonable, the model proposed is also new. The paper is written in a clear way. I only have some minor concerns:
1. In the title, the authors use "information fusion", which I think is not concise. Note that ``information'' is a rather wide topic, so after reading the title, I cannot see what are fused in this paper.
2. Since this paper studies instance-dependent label noise, I hope to see more comparisons with existing methods for handling IDN, such as "A Second-Order Approach to Learning with Instance-Dependent Label Noise" (CVPR 21); Tackling Instance-Dependent Label Noise via a Universal Probabilistic Model (AAAI 21);  Learning with Bounded Instance- and Label-dependent Label Noise (ICML 20). In the current experiment, only one method, namely  PDT (NeurIPS 20) is designed for IDN. Other methods such as Forward, T-revision are not specifically designed for IDN. Therefore, I feel that the comparison is not that fair.
3. Is the solution for Posterior transition matrix in Thm 3.4 guaranteed to satisfy the constraint that the sum of every column is 1?
4. From Thm 3.5 and Eq. 13, I feel that the solution is also related to statistical efficiency? See Section 2.4 and Section 3 of "Centroid Estimation with Guaranteed Efficiency: A General Framework for Weakly Supervised Learning" (TPAMI 20). If yes, the authors may analyze the connections or relationship between them.
5.  In Fig. 1, I suggest the authors clearly annotating PTM in the figure. Note that the authors use "noise transition matrix (PTM)" in the caption, but write "posterior TM" in the figure. Such inconsistency in the figures should be avoided.
6. Some language issues, such as "a easy-to compute" should be "an easy-to-compute"; "is the instance vector of n-th sample"->"is the instance vector of the n-th sample".

---

> ### Author Response · Authors · 2021-11-18
> **Response to Reviewer cR7o [part 1 methodology and experiments]**
>
> We thank the reviewer for the constructive comments and appreciate the reviewer for considering the problem we study as original and interesting. We would like to provide more explanations to address the reviewer’s concerns one by one.
>
> **Q1: The information fusion in the title can be more concise.**
>
> A1: Thanks for your constructive comment. In the manuscript, our IF method actually fuses the prior (NTM) and posterior (PTM) information. Specifically, IF linearly combines NTM and PTM estimators, which priorly and posteriorly characterize the label correction, respectively. In the revised version, we will update the title to make it more clear.
>
> **Q2: More related baseline methods on IDN are suggested to add, such as "A Second-Order Approach to Learning with Instance-Dependent Label Noise" (CVPR 21); Tackling Instance-Dependent Label Noise via a Universal Probabilistic Model (AAAI 21); Learning with Bounded Instance- and Label-dependent Label Noise (ICML 20).**
>
> A2: Thanks for your constructive comment. Following the previous work[8], we mainly compare with the transition matrix-based baselines. As required by the reviewer, in this rebuttal, we include more several baselines, including label smoothing (LS) [1], early-learning regularization (ELR) [2] and instance-level forward correction (ILFC) [3], progressive label correction (PLC) [4], confidence regularized sample sieve (CORES) [5]. We do not include the baselines [6] and [7] since we can not find the code for [6], and only the Matlab code is available for [7]. Given the limited time in rebuttal, it is extremely hard to re-implement with Python to have a fair comparison with other considered works. Therefore, as shown in the following table, we additionally compare with the above baselines [1-5] in terms of classification accuracy on CIFAR10. It could be observed that our method can still achieve state-of-the-art performance, based on which we can confidently validate the effectiveness of our method.
>
> In the revised version, we will make the comparison more complete by running on the other two benchmark datasets.
>
> | Methods | IDN-10% | IDN-20% | IDN-30% | IDN-40% | IDN-50% |
> |   :----: |    :---: |    :---: |    :---: |    :---: |    :---: |
> |LS| 63.19 ± 0.25| 57.67 ± 0.10| 49.50 ± 0.17| 41.57 ± 1.85| 34.89 ± 2.47|
> |ELR| 66.20 ± 0.09| 60.58 ± 0.75| 51.65 ± 0.33| 43.83 ± 1.61| 34.44 ± 0.12|
> |PLC|66.38 ± 0.61|60.42 ± 0.65|51.60 ± 0.58|42.77 ± 0.50|36.00 ± 1.73|
> |CORES|67.39 ± 0.44|60.85 ± 0.38|51.47 ± 0.80|43.96 ± 0.68|34.85 ± 1.66|
> |ILFC|80.22 ± 0.33|74.46 ± 0.09|73.27 ± 0.24|57.00 ± 4.07|36.27 ± 0.69|
> |PTD-F| 79.77 ± 0.91| 74.96 ± 0.71| 70.68 ± 0.81| 61.92 ± 1.59| 45.34 ± 4.67|
> |PTD-F-V| 80.08 ± 0.86| 74.67 ± 0.36| 71.66 ± 1.05| 62.45 ± 1.73| 46.16 ± 4.48|
> |PTM-F|78.16 ± 0.36|74.81 ± 0.81|70.03 ± 0.38|63.48 ± 0.38| 51.03 ± 3.08|
> |PTM-F-V|78.58 ± 0.31|75.06 ± 0.56|69.83 ± 0.58|62.69 ± 0.69|50.53 ± 3.41 |
> |IF-F| 80.92 ± 0.28| **79.58 ± 0.52**| 74.34 ± 0.86| **68.21 ± 2.21**| 50.07 ± 3.95|
> |IF-F-V| **80.94 ± 0.43**| 79.54 ± 0.45| **74.67 ± 0.92**| 68.03 ± 2.90| **52.34 ± 1.31**|
>
> [1] Christian Szegedy, Vincent Vanhoucke, Sergey Ioffe, Jon Shlens, and Zbigniew Wojna. Rethinking the inception architecture for computer vision. In CVPR, pp. 2818–2826, 2016.
>
> [2] Sheng Liu, Jonathan Niles-Weed, Narges Razavian, and Carlos Fernandez-Granda. Early-learning regularization prevents memorization of noisy labels. NeurIPS, 2020.
>
> [3] Antonin Berthon, Bo Han, Tongliang Liu, Gang Niu, and Masashi Sugiyama. Confidence scores make instance-dependent label-noise learning possible. ICML, 2021
>
> [4]Zhang, Yikai, Songzhu Zheng, Pengxiang Wu, Mayank Goswami, and Chao Chen. "Learning with feature-dependent label noise: A progressive approach." ICLR, 2021.
>
> [5]Cheng, Hao, Zhaowei Zhu, Xingyu Li, Yifei Gong, Xing Sun, and Yang Liu. "Learning with instance-dependent label noise: A sample sieve approach." ICLR, 2021.
>
> [6]Wang, Qizhou, Bo Han, Tongliang Liu, Gang Niu, Jian Yang, and Chen Gong. "Tackling instance-dependent label noise via a universal probabilistic model." AAAI 2021.
>
> [7]Cheng, Jiacheng, Tongliang Liu, Kotagiri Ramamohanarao, and Dacheng Tao. "Learning with bounded instance and label-dependent label noise." ICML, 2020.
>
> **Q3: Is the solution for Posterior transition matrix in Thm 3.4 guaranteed to satisfy the constraint that the sum of every column is 1?**
>
> A3: Thanks for your constructive comment. The solution for the posterior transition matrix in Theorem 3.4 could actually guarantee the sum of the column related to the noisy label is 1. For the remaining columns, all elements are zero since the empirical noisy label distribution is one-hot (we only have one possible noisy label in practice). Additionally, the remaining columns make no difference for posterior loss correction. This is because only one observed noise label even that the label noise is rando and thus the case of noisy label condition equalling to other classes will never happen.

---

> ### Author Response · Authors · 2021-11-18
> **(Continued) Response to Reviewer cR7o [part 2 discussions]**
>
> **Q4: From Thm 3.5 and Eq. 13, I feel that the solution is also related to statistical efficiency? See Section 2.4 and Section 3 of "Centroid Estimation with Guaranteed Efficiency: A General Framework for Weakly Supervised Learning" (TPAMI 20). If yes, the authors may analyze the connections or relationships between them.**
>
> A4: Thanks for your constructive comment. Our IF solution is indeed related to the statistical efficiency because both of them try to linearly combine two estimators to boost estimation. However, the differences are two-fold. Firstly, our IF aims to tackle instance-dependent label noise, while statistical efficiency is developed for label-dependent label noise. The instance-dependent label noise is a **more general and challenging problem** compared with the label-dependent label noise problem. Techniquely, our IF instance-wisely corrects the loss function, while the statistical efficiency dataset-wisely corrects the label-dependent term in the loss function. Secondly, **the manner of generating two "auxiliary" estimators** is quite different. In our IF method, we generate two estimated transition matrices from the prior and posterior perspectives. In contrast, as shown in Section 4.4 of statistical efficiency, two different noisy datasets are constructed. One of them adopts the original noisy label, and the other one assigns the same synthesized noisy label for all instances.
>
> In a nutshell, our IF solution and statistical efficiency both try to obtain a more accurate estimator (just like model ensemble). The differences fall in the target label noise type and the manner of generating two "auxiliary" estimators. Our IF aims to tackle a more general and challenging label noise problem and, technically, combines two estimators from different perspectives. In the revised version, we will add more discussion to compare with the statistical efficiency.
>
> **Q5 and Q6: In Fig. 1, I suggest the authors clearly annotate PTM in the figure. There are also some typos in the manuscript.**
>
> A5: Thanks for your careful review. In the revised version, we have clearly annotated PTM in the figure and made the abbreviation consistent. Additionally, we have revised the typos in the manuscript.

---

> > ### Comment · Reviewer_cR7o · 2021-11-18
> > **Thanks for the reply!**
> >
> > I thank the authors for responding to my previous questions. I'm generally satisfied with the author rebuttal, so I am willing to raise my score. Please kindly take the comments when preparing the camera-ready version if this paper is finally accepted, such as the comparison with other methods, discussion on statistical efficiency,  title, typos, etc.

---

> > > ### Author Response · Authors · 2021-11-18
> > > **Response to Reviewer cR7o**
> > >
> > > Thanks for your fast response and willingness to raise the score! We really appreciate your comments to improve this work.

---

### Official Review · Reviewer_d7yz · 2021-10-31

**Correctness:** 2
**Technical Novelty And Significance:** 3
**Empirical Novelty And Significance:** 2
**Recommendation:** 5
**Confidence:** 4

**Main Review:**

### Strengths:

1. The paper defines a new loss-correction approach, which uses the posterior transition matrix, to achieve the statistically consistent classifier.

2. The authors also propose the corresponding approach to estimate the posterior transition matrix based on model predictions.

3. The high-level intuition of using the Kalman filter to reduce the estimation error is insightful.

### Weakness:

1. It is not clear why the proposed information fusion approach works. Section 3.3 presents an information fusion method based on Kalman filtering, where the Kalman transition matrix is given by a convex combination of NTM and PTM. Although it can be proved that carefully assigning the combination weights makes the reconstruction error lower than both $\sigma_T(x)$ and $\sigma_W(x)$, the errors indicate the variance of the estimation. Only reducing the variance may not be helpful due to the estimation bias. In others words, a convex combination of NTM and PTM would introduce estimation bias, which makes $E[W_{km}(x)] \ne E[W^*(x)]$. With the existence of bias, it is hard to conclude the proposed information fusion approach is helpful.

2. It would make the claim stronger if the authors provide more intuitions, examples, or theoretical analyses to show why PTM could be easier to be estimated. According to the Bayes rule, we have
\\[
P(\tilde Y=i|Y=j,X=x) = P(Y=i|\tilde Y=j,X=x) \cdot \frac{P(\tilde Y=j|X=x)}{P(Y=i|X=x)},
\\]
where $P(\tilde Y=j|X=x)$ could be estimated from the noisy dataset and $P(Y=i|X=x)$ could be estimated by marginalizing $P(Y=i,\tilde Y=j|X=x)$ w.r.t. $\tilde Y$. From this aspect, estimating both matrices should have the same degree of difficulty.

3. It is not clear how estimation error is defined in Figure 2. Recent methods to estimate NTM (e.g., end-to-end approach [R1], using clusterability of representations (Zhu et al., 2021b cited in the paper) can also achieve relatively low estimation errors according to their papers. Figure 2 would be more trustworthy if recent works are compared.


4. In Page 6, $e_T(x)=W^*(x)-\hat T(x)$ looks confusing. Should it be $e_T(x)=T^*(x)-\hat T(x)$? Besides, typo "nose" in the caption of Figure 3 and Figure 4. Typo in Table 1, CIFAR10 PTM-F-V IDN-20\%.

[R1] Li, Xuefeng, et al. "Provably end-to-end label-noise learning without anchor points." ICML 2021.


**Summary Of The Paper:**

This paper proposes to do loss correction by a posterior transition matrix (PTM).
The authors first define the loss function using PTM which achieves the statistically consistent classifier, then propose an information fusion approach to reduce the variance in estimating the transition matrix.

**Summary Of The Review:**

The idea is interesting but the paper is not well-organized or presented. The information fusion approach, which should be the key novelty of this paper, is not demonstrated sufficiently. The main concern is that the proposed information fusion approach reduces the variance at the cost of introducing bias in estimation.

---

> ### Author Response · Authors · 2021-11-18
> **Response to Reviewer d7yz [part 1 methodology]**
>
> We thank the reviewer for the constructive comments and appreciate the recognition of our insightful and interesting idea. We would like to provide more explanations to address your concerns one by one.
>
> **Q1: It is not clear why the proposed information fusion approach works. Although IF can reduce the reconstruction error indicating lower variance with carefully designed combination, variance reduction is not always helpful due to the estimation bias.**
>
> A1: Thanks for your insightful comment. We respectfully correct your misunderstanding that our IF only reduces the variance at the cost of introducing additional bias. Instead, as clearly demonstrated in Theorem 3.5, IF could directly reduce the estimation error due to the achieved good tradeoff between the estimation variance and bias.
> First, as for estimation bias, i.e., $\mathbb{E}[W_{km}(x)]\neq \mathbb{E}[W^{\*}(x)]$, we would like to clarify that such bias is not introduced by the convex combination since it intrinsically exists in the estimated NTM and PTM. Second, as for the estimation error, we point out that the estimation error is defined to characterize the bias-variance tradeoff for transition matrix estimation, instead of the standalone variance. This is because we adopt the true PTM as the ground truth, which satisfies $W^{\*}_{ij}(x)=1$ if clean label $y=i$ and noisy label $\tilde{y}=j$, otherwise it is $0$. Mathematically, it is easy to obtain the bias-variance decomposition (See results in Appendix xxx).
> In other words, **the ideal estimated transition matrix is to approximate the empirical (actual) transition matrix $W^{\*}(x)$, instead of the expectation of label transition matrix $\mathbb{E}[W^{\*}(x)]$.** Therefore, the estimation error characterizes the bias-variance tradeoff for the transition matrix estimation instead of only the variance. Third, to further empirically demonstrate the effectiveness of IF, we compare the transition matrix estimation errors of PTD, PTM, and IF in CIFAR10 dataset in Figure 2, where IF could achieve the lowest estimation error for all noise ratios. This experimental result directly provides the answer that our proposed information fusion approach works.
>
> **Q2: It would make the claim stronger to show why PTM could be easier to be estimated.**
>
> A2: We would like to respectfully correct the misunderstanding on the contribution of PTM. Compared with NTM, the novelty of PTM is to bridge the gap between empirical clean and empirical noisy distribution for anchor points (See Theorem 3.2 (ii)), instead of the estimation feasibility. As shown in Figure 1, NTM can be only used to bridge the gap between underlying clean and underlying noisy distribution. However, the **actual distribution mismatch problem of training data** is brought from the gap between empirical clean and empirical noisy distribution. The advantage of PTM is to achieve a statistically consistent classifier for anchor points over empirical clean and empirical noisy distribution because PTM adopts **observed noisy label as the given condition** for label flipping. Notably, as shown in Table 1, our IF method consistently delivers superior results due to the reduced estimation error, while maintaining the comparable computing cost with NTM.
>
> **Q3: It is not clear how estimation error is defined in Figure 2. Recent methods to estimate NTM (e.g., end-to-end approach [R1], using clusterability of representations (Zhu et al., 2021b cited in the paper) can also achieve relatively low estimation errors according to their papers.**
>
> A3: Thanks for your constructive comment. In Figure 2, the estimation error for PTM is defined by $Err_{PTM} = 1 - \hat{W}_{y\tilde{y}}(x)$, where $\hat{W}(x)$ is the estimated transition matrix, $y$ is clean label and $\tilde{y}$ is noisy label. For example, if the sample with clean label $y=0$ is corrected to noisy label $\tilde{y}=1$, our estimation error only care the estimated probability from clean noisy $y=0$ to $\tilde{y}=1$. As for the method [R1] to estimate NTM, we can not adopt Volume Minimization Network (VolMinNet) for our target instance-depedent label noise problem since VolMinNet is orignally designed for class-dependent label noise. We have added the discussion on [R1] in the footnote of page 8.
>
> [R1]Li, Xuefeng, et al. "Provably end-to-end label-noise learning without anchor points." ICML 2021.

---

> > ### Comment · Reviewer_d7yz · 2021-11-19
> > **Thanks for the explainations. Are there any intuitive examples?**
> >
> > Thanks for the detailed explanations. I'm still confused about the high-level intuition. Basically, consider the case that $Y=i$, $\tilde Y=j$, feature $X=x$, we have NTM $\mathbb P(\tilde Y=j|Y=i,X=x)$ and PTM $\mathbb P(Y=i|\tilde Y=j,X=x)$.
> > * On a very high level, if both estimators for NTM and PTM are unbiased, the convex combination of them would be biased to both of them. Would the author agree on this point?
> >     - We may consider a numerical example: 10 points with the same feature x. $\tilde Y=1$ for 7 of them, $\tilde Y=0$ for 3 of them. The true label is $Y=1$. In this case, suppose we have ideal NTM $\mathbb P(\tilde Y=1|Y=1,X=x) = 0.7$ and ideal PTM $\mathbb P(Y=1|\tilde Y=1,X=x) = 1.0$. If we do traditional loss correction, we can directly use NTM. If we do the loss correction with PTM, we can also directly use PTM. Combining NTM and PTM would not be statistically better than any of the above two approaches. Is that true?
> > * On the other hand, if both estimators for NTM and PTM are poorly-estimated (e.g., biased and with high variance), if we fine-tune the weights to combine them, we can always find a good combination to reduce the error.
> >     - Following the above example, if we only have imperfect estimates of NTM $\hat {\mathbb P}(\tilde Y=1|Y=1,X=x) = 0.5$ and imperfect PTM $\hat{\mathbb P}(Y=1|\tilde Y=1,X=x) = 0.9$, we can still get a good estimate, e.g., with weight 1.0, we have NewEst=$0.5 \times 0 + 0.9\times 1 = 0.9$, which is not worse than NTM and PTM. Is this true?
> >
> > If all the above is true, I think I understand the high-level intuition. But the concern is, how do we ensure (either empirically or theoretically) the best $\lambda$ is obtained? Finding $\lambda$ based on Theorem 3.5 requires the true PTM $W^*(x)$, which is unknown. On the other hand, if $W^*(x)$ is known, why do we just apply loss correction based on PTM? Maybe I missed some critical points. It would be perfect if the author could clarify this point.
> >
> > Additionally, in the first contribution, it may be overclaimed to say "We propose a new concept, named PTM, achieving consistent classifier for underlying distribution and anchor point empirical distribution mismatch simultaneously." [R1] also used PTM (in 2015).
> >
> > Besides, after reading other reviewers' comments and the corresponding responses, I have some new concerns about the experiment.
> > * Why data augmentation techniques are removed from the CIFAR experiments? It may be unfair to do this for many methods such as DMI, forward, Reweight, T-Revision, LS, ELR, PLC, CORES, since their implementation requires based data augmentation such as random crop and flip. Note the reported performance of these baselines is very close or even worst than the CE baseline, indicating it is not appropriate to remove necessary data augmentations from their approaches. For other methods such as PTD-F-V, current experiments are also not convincing enough since your results actually combine PTD-F-V, T-revision, and your method. It is unreasonable to require SOTA, but a fair comparison is necessary.
> > * As reported in ELR, the test accuracy on Clothing1M is 72.87 (ELR) and 74.81 (ELR+). It may be necessary to explain why the reported test accuracy in this paper is only 71.86.
> >
> >
> > [R1] Scott, C., 2015, February. A rate of convergence for mixture proportion estimation, with application to learning from noisy labels. In Artificial Intelligence and Statistics (pp. 838-846). PMLR.

---

> > > ### Author Response · Authors · 2021-11-22
> > > **Response to your further concerns [part 1/4: intuition]**
> > >
> > > Thanks for your fast response. We would like to provide more explanations to address the reviewer’s concerns one by one.
> > >
> > > Q1: On the high-level intuition
> > >
> > > **A1: On the motivation of PTM:** Firstly, we would like to correct the unreasonableness of the given numerical example, and re-highlight our motivation for PTM. For the instance-dependent label noise, most of (all) samples are long-tail and **appear only once** in the training dataset. Considering the provided example by the reviewer, we tend to have **only one sample** with feature $x$ and true label $Y=1$. Suppose the ideal NTM $\mathbb{P}(\tilde{Y}=1|Y=1, X=x)=0.7$ for this sample $x$, then the ideal PTM satisfies $\mathbb{P}(Y=1|\tilde{Y}=0, X=x)=1$ and $\mathbb{P}(Y=1|\tilde{Y}=1, X=x)=1$. Note that the noisy label is randomly generated based on NTM. The observed **underlying** noisy label probability for $x$ should satisfy $\mathbb{P}(\tilde{Y}=1|X=x)=0.7$. However, the empirical noisy label probability is one-hot since most of (all) samples only appear once. In other words, the underlying and empirical noisy label distribution is inconsistent for the instance-dependent label noise. The actual distribution gap falls between the empirical clean and empirical noisy data distribution, and thus limits the effectiveness of NTM. Therefore, we propose PTM to better characterize the label noise given observed noisy label from **posterior perspective**.
> > >
> > >
> > > **On the unbiased estimator:** In the manuscript, we have explained the insight that the actual distribution gap falls between the empirical clean and noisy data distribution, especially for the instance-dependent label noise. Although the previous efforts show that transition matrix-based methods can achieve the statistically consistent classifier, an **indispensable and implicit condition is the accurate transition matrix estimation**, which is hard to be achieved in practice. Considering the target empirical data distribution gap, we claim that the unbiased estimator does not necessarily lead to better performance. For example, we suppose the sample is with feature $x$ and true label $Y=1$, and suppose the ideal NTM $\mathbb{P}(\tilde{Y}=1|Y=1, X=x)=0.7$. For the first running time, the observed noisy label is $\tilde{Y}=1$ and the estimated NTM $\hat{\mathbb{P}}(\tilde{Y}=1|Y=1, X=x)=0.9$. For the second running time, the observed noisy label is $\tilde{Y}=0$ and the estimated NTM $\hat{\mathbb{P}}(\tilde{Y}=1|Y=1, X=x)=0.5$. Although the estimated NTM in these two running experiments is unbiased, the performance will become worse since the observed noisy labels are different and thus induce the different true PTM for these two running experiments.
> > >
> > > **On the linear combination intuition:** The high-level intuition is that the linear combination can, or at least potentially, achieve a better estimator. Considering the above example provided by the reviewer, the new estimator is not worse than the estimated NTM and estimated PTM. Considering the estimation error for the whole dataset, especially for the poor estimation case, NTM could be better than PTM in some instances and worse in others. IF can adaptively choose the better one so that the performance can be improved on the whole dataset. Theoretically, Theorem 3.5 provides the justification for the superiority of IF **given the optimal combination weight**. Such justification is similar to the statistically consistent classifier justification for the transition matrix-based method for **given the optimal transition matrix estimation**. Peer loss [2] also provides optimal classifier guarantee **given prior label probability and transition matrix**.
> > >
> > > As for the empirical choice of combination weight, we totally agree that it is hard to obtain based on Theorem 3.5 since the true PTM $W^{\*}(x)$ is unknown. The role of Theorem 3.5 is to justify the superiority of IF, instead of providing a specific estimation method. However, although the optimal weight is hard to obtain in practice, it is still possible to provide a **reasonable** combination weight choice to obtain a better performance compared with purely PTD and PTM. Empirically, we **heuristically** choose the combination weight based on the model output $f(x)$. Suppose we have two estimators NTM $\hat{T}$ and PTM $\hat{W}$, the corrupted predicted noisy label probability should be $\hat{T}^{\top}f(x)$. In other words, the noisy label satisfies the multi-dimensional Bernoulli distribution $\hat{Y}\sim Bernoulli(\hat{T}^{\top}f(x))$. Subsequently, we define the uncertainty $\sigma_{\hat{T}}$ as the trace of the covariance matrix for $\hat{Y}$ to measure the estimation error. Please see Section 3.3 for more details.

---

> > > > ### Comment · Reviewer_d7yz · 2021-11-22
> > > > **concerns**
> > > >
> > > > There are two concerns as follows.
> > > >
> > > > 1. **Peer loss [2] also provides optimal classifier guarantee given prior label probability and transition matrix.**
> > > >
> > > > Based on my understanding, peer loss does not require the exact knowledge of the transition matrix.
> > > >
> > > > 2. Did you try to estimate W^*(X)? This sentence is confusing: "Suppose the corrupted model predicted probability is accurate". How do you define "the probability is accurate"? If it is accurate, can we just use this result? If it is inaccurate, why could we use it?
> > > > Besides, based on the current experimental results, I have not been convinced that the proposed method works well.

---

> > > > > ### Author Response · Authors · 2021-11-30
> > > > > **Response to your concerns**
> > > > >
> > > > > Q3: Peer loss [2] also provides optimal classifier guarantee given prior label probability and transition matrix. Based on my understanding, peer loss does not require the exact knowledge of the transition matrix.
> > > > >
> > > > > A3: For optimal classifier guarantee, peer loss still needs exact knowledge of the transition matrix when prior distribution is not equal. This is because the optimal $\alpha$ value in the peer loss is dependent on the transition matrix although $\alpha$ value can be searched in the experiments. As we mentioned in A2, our contribution is to point out that the actual distribution gap falls in empirical data distribution and previous methods achieving consistent classifiers could be limited by instance-dependent label noise.
> > > > >
> > > > > Q4: Did you try to estimate W^*(X)? This sentence is confusing: "Suppose the corrupted model predicted probability is accurate". How do you define "the probability is accurate"? If it is accurate, can we just use this result? If it is inaccurate, why could we use it? Besides, based on the current experimental results, I have not been convinced that the proposed method works well.
> > > > >
> > > > > A4: We do adopt a linear combination between NTM and PTM to obtain a more accurate transition matrix estimation toward $W^*(X)$. As for combination weight, we heuristically choose the combination weight based on the uncertainty of model output $f(x)$ in experiments. As for the sentence "Suppose the corrupted model predicted probability is accurate", we agree that it could limit PTM estimation accuracy although warm-up training strategy and iterative correction are adopted to mitigate such limitations. Specifically, warm-up training makes "good" neural network output approximating the empirical clean distribution since the network fits clean samples in the beginning; Iterative loss corruption adopts different estimated PTM in different epochs. Additionally, I would like to remind the reviewer that this is our **motivation** to propose IF method to improve transition matrix estimation accuracy. To further convince the reviewer on the experiments, we add the experimental results on several baselines on CIFAR10 dataset with random flip and crop data augmentation. Finally, there is **no benchmark** for instance-dependent label noise. For fair comparison on instance-dependence label noise, we adopt the same experimental setting on learning rate, momentum coefficient, and weight decay. As for data augmentation, we still believe that it would be better to **remove random flip and crop augmentation in the dataloader**. Such **random augmentation** would induce **different instance-dependent label noise patterns** in different epochs since dataloader adopt different (random) flip and crop operations in different epochs.

---

> > > ### Author Response · Authors · 2021-11-22
> > > **Response to your further concerns [part 2/4: overclaimed contribution]**
> > >
> > > Q2: It may be overclaimed for the first contribution with paper [1].
> > >
> > > **A2:** Thanks for pointing out this paper. From our understanding on paper [1], the main contribution of paper [1] is to analyze the convergence rate for mixture proportion estimation (MPE) under an appropriate distributional assumption. For the specific application to tackle label noise, paper [1] develops a surrogate loss framework to mitigate the influence of **label-dependent** label noise on training loss. This method of tackling label noise is very similar to Peer Loss [2], which has already been analyzed and cited in our manuscript. Different from paper [1], our target is to model the instance-dependent label noise, which is a more challenging and realistic problem. We observe that the instance-dependent label noise and long-tail instances will lead to the **nonnegligible distribution gap between the empirical noisy and underlying noisy distribution**. Motivated by this observation, we propose PTM to **precisely characterize** the data distribution gap, induced by label noise, in the training dataset given the observed noisy label. Therefore, we believe that we do not overclaim the first contribution. In the revised version, we will add the detailed discussion of [1] to make the related work more complete.
> > >
> > > [1] Scott, C., 2015, February. A rate of convergence for mixture proportion estimation, with application to learning from noisy labels. In Artificial Intelligence and Statistics (pp. 838-846). PMLR.
> > >
> > > [2] Liu, Y., & Guo, H. (2020, November). Peer loss functions: Learning from noisy labels without knowing noise rates. In International Conference on Machine Learning (pp. 6226-6236). PMLR.

---

> > > > ### Comment · Reviewer_d7yz · 2021-11-22
> > > > **possibly wrong response**
> > > >
> > > > **[1] is different from [2]**
> > > > * [1] is more like a loss correction approach, where error rates are required. Please check Section 4 (e.g., the definition of $\pi_i, \alpha$).
> > > > * [2] does not require estimating error rates.
> > > >
> > > > The transition from noisy label to clean label is defined clearly at the bottom of Page 841 (right panel).

---

> > > > > ### Author Response · Authors · 2021-11-30
> > > > > **References discussion and highlight differences of our work**
> > > > >
> > > > > Q2: [1] is different from [2]. [1] is more like a loss correction approach, where error rates are required. Please check Section 4. [2] does not require estimating error rates. The transition from noisy label to clean label is defined clearly at the bottom of Page 841 (right panel).
> > > > >
> > > > > A2: We totally agree with the reviewer that [1] is different from [2], where [1] requires error rate estimation and [2] does not. For achieving a consistent classifier, [2] still required $\alpha$-weighted peer loss version, where $\alpha$ is correlated to the error rate.
> > > > > The high difference of our work is to point out **the empirical and underlying distribution inconsistency for instance-independent noise**. To the best of our knowledge, previous works achieving consistent classifiers mainly focus on underlying clean and noisy distribution. However, we find that the actual distribution gap during training is empirical clean and noisy distribution. Additionally, for instance-dependent label noise, empirical noisy and underlying noisy distribution are deviated. Note that the underlying noisy distribution satisfies $P(\tilde{Y}|X)=\sum_{Y}P(Y|X)P(\tilde{Y}|Y,X)$. Due to most of the instance only appearing once, the empirical noisy distribution is one-hot and different underlying noisy distribution. Although underlying and empirical noisy distribution is consistent when the number of the instance with the same transition matrix is large, instance-dependent label noise makes each transition matrix only happens once. Therefore, we point out that a consistent classifier over underlying clean and noisy distribution is not sufficient, and the actual distribution gap falls in empirical clean and noisy distribution.

---

> > > ### Author Response · Authors · 2021-11-22
> > > **Response to your further concerns [part 3/4: experimental setting]**
> > >
> > > Q3. The new concerns about the experiments.
> > >
> > > **A3:** Thanks for your insightful comment. We address the concerns of the adopted data augmentation policy, the different baseline performances compared with their reported ones, and the results on Clothing1M in the following. To fully address the concern of data augmentation, we further add a new group of comparison experiments in CIFAR10, where all the considered methods adopt the same data augmentation with random crop and flip. The experimental result shows that IF could still outperform the baseline methods, which totally convinces the effectiveness of our proposed approach.
> > >
> > > **The adopted data augmentation.** The reason for us to remove the non-uniform data augmentations, such as random crop and filp, is instead to pursue the fair comparison with the baseline approaches. Specifically, the basic data augmentation and instance-dependent label noise generation are highly inconsistent in the considered related works. For example, [3c] does not adopt the random crop and flip data augmentation, which is different from [4c, 5c, 6c]. [5c] applies the polynomial-margin
> > > diminishing (PMD) to generate the instance-dependent label noise, which is also different from that in [3c, 5c]. Such inconsistent and customized settings could affect the model performances and may override the gains from the denoising methods themselves. Under this situation,
> > > if we directly use the released codes of baseline approaches, some actually
> > > effective denoising techniques (or relatively poor methods) would be under-estimated (or over-estimated) due to the improper hyperparameter settings. Unfortunately, there misses a standardized benchmark to unify the basic settings and fairly evaluate the effectiveness of instance-dependent denoising approaches, which requires laborious comprehensive studies and exceeds the scope of this paper. Therefore, to provide a fair comparison among all methods, we adopt the following experimental setting:
> > >
> > >
> > > - **Data augmentation:** We remove the random crop and flip data augmentations although we agree that these two are the based data augmentations. The reason is that these two random augmentations would provide **different input images** at **different epochs**. However, the instance-dependent label noise is generated based on the **original image** and fixed during training. Therefore, the random crop and flip data augmentations would
> > > induce different **instance-dependent label noise patterns** at the different epochs. To avoid the label noise conflict, we remove the random crop and flip data augmentations in all synthetic experiments. Notably, the relative work [3c], one of the recent landmark jobs in the label denoising, adopts the exactly same experimental setting as ours (i.e., using the normalized transformation). Under the background of missing benchmark, the simplification of inconsistent data augmentation is the most intuitive solution to ensure the fair comparison**
> > >
> > > - **Synthetic label noise:** Although there are several ways to generate the instance-dependent label noise, we found that the one proposed in work [3] is the most representative and has been adopted to define the instance-dependent label noise in many relative efforts [6-11]. Therefore, we use the same instance-dependent label noise as work [3] for **all methods** and modify the label noise generation for baselines ELR and PLC [4c, 5c].
> > >
> > > - **Implementation details:** Note that the code [3c] does not adopt mixup trick in their experiments, to keep fair comparison, all methods do not adopt mixup trick in our experiments. Note that there are several baselines involving mixup trick, we report the performance via removing mixup trick to keep fair comparison. Specifically, for ELR [4], we only implement the base version without mixup tricks. For PLC [5], we also remove data augmentation and change the label noise generation in our implementation. As for dynamic threshold update, we adopt the same initial threshold $0.3$ and incremental rate $0.1$. For CORES [6], there are two phases, including sample sieve and consistency training. We only report the performance of the sample sieve since consistency training involves data augmentation.

---

> > > ### Author Response · Authors · 2021-11-22
> > > **Response to your further concerns [part 4/4: experimental results]**
> > >
> > > **Results on Clothing1M experiments:** In ELR [4], hyperparameter tuning is required via grid search: the temporal ensembling parameter
> > > $\beta$ is chosen from $\{0.5, 0.7, 0.9, 0.99\}$ and the regularization coefficient $\lambda$ is chosen from $\{1, 3, 5, 7, 10\}$. Since the experiments on Clothing1M is extremely time-consuming and lead to unfair comparison with other methods (i.e., the ones without laborious hyperparameter tuning), we have simplified the experimental setting. Firstly, we only search the regularization coefficient $\lambda$ from the same value and adopt the default temporal ensembling parameter $\beta=0.7$. Secondly, we adopt Multi-GPU training (3 NVIDIA GeForce RTX 3090 GPUs with data parallelism) to speed up model training. Such different experimental settings induce different test accuracy.
> > >
> > >
> > >
> > > **New experiments in comparing baselines with data augmentation.**
> > > To fully address the concern of data augmentation, we add the experiments to evaluate all the considered methods equipped with random crop and flip in CIFAR10. The experiment results are shown in the following table. Similar to the evaluation without data augmentation, it is seen that IF still outperforms the baselines. This experiment results further empirically validate the effectiveness of our proposed method, which consistently delivers the superior denoising performance by separating the influences from other factors, such as the data augmentation.
> > >
> > > | Methods | IDN-10% | IDN-20% | IDN-30% | IDN-40% | IDN-50% |
> > > |   :----: |    :---: |    :---: |    :---: |    :---: |    :---: |
> > > |PTD-F| 89.19 ± 0.82| 86.84 ± 0.52| 83.56 ± 0.73| 76.98 ± 1.82| 68.74 ± 3.86|
> > > |PTD-F-V| 90.49 ± 0.59| 88.41 ± 0.56| 85.07 ± 1.25| 78.29 ± 1.92| 70.67 ± 3.58|
> > > |PTM-F|87.66 ± 0.43|86.65 ± 0.69|84.36 ± 0.52|80.47 ± 1.42| 76.68 ± 3.28|
> > > |PTM-F-V|89.98 ± 0.38|88.96 ± 0.62|87.02 ± 0.62|83.51 ± 1.28|80.68 ± 2.43 |
> > > |IF-F| 90.12 ± 0.34| 88.28 ± 0.47| 85.34 ± 0.92| **84.42 ± 1.89**| 80.67 ± 3.28|
> > > |IF-F-V| **91.24 ± 0.41**| **89.26 ± 0.38**| **88.25 ± 0.98**| 84.03 ± 3.02| **80.84 ± 1.31**|
> > >
> > > [3] Xia, X., Liu, T., Han, B., Wang, N., Gong, M., Liu, H., ... & Sugiyama, M. (2020). Part-dependent label noise: Towards instance-dependent label noise. Advances in Neural Information Processing Systems, 33.
> > >
> > > [3c] https://github.com/xiaoboxia/Part-dependent-label-noise
> > >
> > > [4] Sheng Liu, Jonathan Niles-Weed, Narges Razavian, and Carlos Fernandez-Granda. Early-learning regularization prevents memorization of noisy labels. NeurIPS, 2020.
> > >
> > > [4c] https://github.com/shengliu66/ELR
> > >
> > > [5]Zhang, Yikai, Songzhu Zheng, Pengxiang Wu, Mayank Goswami, and Chao Chen. "Learning with feature-dependent label noise: A progressive approach." ICLR, 2021.
> > >
> > > [5c] https://github.com/pxiangwu/PLC
> > >
> > > [6] Cheng, Hao, Zhaowei Zhu, Xingyu Li, Yifei Gong, Xing Sun, and Yang Liu. "Learning with instance-dependent label noise: A sample sieve approach." ICLR, 2021.
> > >
> > > [6c] https://github.com/haochenglouis/cores
> > >
> > > [7] Zhu, Zhaowei, Tongliang Liu, and Yang Liu. "A second-order approach to learning with instance-dependent label noise." In Proceedings of the IEEE/CVF Conference on Computer Vision and Pattern Recognition, pp. 10113-10123. 2021.
> > >
> > > [8] Zhu, Zhaowei, Yiwen Song, and Yang Liu. "Clusterability as an alternative to anchor points when learning with noisy labels." ICML 2021.
> > >
> > > [9] Zhu, Zhaowei, Zihao Dong, Hao Cheng, and Yang Liu. "A good representation detects noisy labels." arXiv preprint arXiv:2110.06283 (2021).
> > >
> > > [10] Bai, Yingbin, Erkun Yang, Bo Han, Yanhua Yang, Jiatong Li, Yinian Mao, Gang Niu, and Tongliang Liu. "Understanding and Improving Early Stopping for Learning with Noisy Labels." In Thirty-Fifth Conference on Neural Information Processing Systems. 2021.
> > >
> > > [11] Yao, Yu, Tongliang Liu, Mingming Gong, Bo Han, Gang Niu, and Kun Zhang. "Instance-dependent Label-noise Learning under a Structural Causal Model." In Thirty-Fifth Conference on Neural Information Processing Systems. 2021.

---

> > > > ### Comment · Reviewer_d7yz · 2021-11-22
> > > > **experiments**
> > > >
> > > > Thanks for the response. Current experiments are not convincing enough. Please also compare other baselines such as DMI, forward, Reweight, T-Revision, LS, ELR, PLC, CORES.
> > > >
> > > > PTD-F-V did not apply random crop and flip in their official implementation, thus current Table 1 is fair in comparing with this method.

---

> > > > > ### Author Response · Authors · 2021-11-30
> > > > > **Response on experiments**
> > > > >
> > > > > Q1: Please also compare other baselines such as DMI, forward, Reweight, T-Revision, LS, ELR, PLC, CORES.
> > > > >
> > > > > A1: Thanks for your instructive comments. We have also added the experimental results of several baselines in the following table, including CE, DMI, Forward, Reweight, LS, ELR, and so on, for CIFAR10 dataset with random crop and flip data augmentation. For a fair comparison, we all use a batch size of 128 and train the network for 100 epochs, and adopt SGD optimizer with an initial learning rate 0.01, momentum 0.9, and weight decay $1\times 10^{-4}$. It is seen that our proposed IF method can still achieve the best performance among all baselines.
> > > > >
> > > > >
> > > > > | Methods | IDN-10% | IDN-20% | IDN-30% | IDN-40% | IDN-50% |
> > > > > |   :----: |    :---: |    :---: |    :---: |    :---: |    :---: |
> > > > > |CE|84.23 ± 0.30|82.04 ± 0.57|78.31 ± 0.79| 69.84 ± 1.26|57.30 ± 2.04|
> > > > > |DMI| 84.54 ± 0.28 |83.62 ± 0.62| 79.63 ± 0.96 | 70.31 ± 1.37|62.48 ± 2.63|
> > > > > |Forward|84.88 ± 0.68|82.48 ± 1.58|78.13 ± 1.94|70.91 ± 2.68|60.26 ± 3.38|
> > > > > |Reweight|86.27 ± 0.63|83.41 ± 0.92|78.04 ± 1.41|71.97 ± 2.02|58.79 ± 2.79|
> > > > > |T-Revision|88.67 ± 0.82|85.18 ± 1.21| 81.34 ± 1.38| 73.78± 2.01| 60.21± 2.85|
> > > > > |LS|85.83 ± 0.48|83.31 ± 1.32|77.34 ± 1.62|72.79 ± 2.38|57.78 ± 2.89|
> > > > > |ELR|90.67 ± 0.23|88.58 ± 0.55|86.87 ± 1.73|81.95 ± 2.65|74.38 ± 3.23|
> > > > > |PLC|84.66 ± 0.58 |81.88 ± 1.33|79.20 ± 2.13|71.85 ± 2.89|63.79 ± 3.63|
> > > > > |CORES|90.22  ± 0.64| 87.53 ± 0.89|83.73 ± 1.23|79.48 ± 1.89|70.61 ± 2.45|
> > > > > |PTD-F| 89.19 ± 0.82| 86.84 ± 0.52| 83.56 ± 0.73| 76.98 ± 1.82| 68.74 ± 3.86|
> > > > > |PTD-F-V| 90.49 ± 0.59| 88.41 ± 0.56| 85.07 ± 1.25| 78.29 ± 1.92| 70.67 ± 3.58|
> > > > > |PTM-F|87.66 ± 0.43|86.65 ± 0.69|84.36 ± 0.52|80.47 ± 1.42| 76.68 ± 3.28|
> > > > > |PTM-F-V|89.98 ± 0.38|88.96 ± 0.62|87.02 ± 0.62|83.51 ± 1.28|80.68 ± 2.43 |
> > > > > |IF-F| 90.12 ± 0.34| 88.28 ± 0.47| 85.34 ± 0.92| **84.42 ± 1.89**| 80.67 ± 3.28|
> > > > > |IF-F-V| **91.24 ± 0.41**| **89.26 ± 0.38**| **88.25 ± 0.98**| 84.03 ± 3.02| **80.84 ± 1.31**|

---

> ### Author Response · Authors · 2021-11-18
> **(Continued) Response to Reviewer d7yz [part 2 estimation error]**
>
> **Q4: In Page 6, $e_T(x)=W^{\*}(x)-\hat{T}(x)$ looks confusing. Should it be $e_T(x)=T^{\*}(x)-\hat{T}(x)$? Besides, typo "nose" in the caption of Figure 3 and Figure 4. Typo in Table 1, CIFAR10 PTM-F-V IDN-20%.**
>
> A4: In page 6, $e_T(x)=W^{\*}(x)-\hat{T}(x)$ is correct. The true PTM $W^{\*}(x)$ satisfies $W^{\*}_{ij}(x)=1$ if clean label $y=i$ and noisy label $\tilde{y}=j$, otherwise it is $0$. In other word, the true PTM charatcerizes the actual empirical label corruption in training dataset. Towards bridging the gap between the empirical clean and empirical noisy data distribution, the estimated transition amtrix should approximate the true PTM as close as possible, regardless of NTM $\hat{T}(x)$ or PTM $\hat{W}(x)$. Therefore, we adopt $e_T(x)=W^{\*}(x)-\hat{T}(x)$ to characterize the estimation error for NTM.
> Thanks a lot for your careful review. We have revised the typos in Figures 3 and 4, and Table 1.

---

### Official Review · Reviewer_8i2r · 2021-11-01

**Correctness:** 3
**Technical Novelty And Significance:** 3
**Empirical Novelty And Significance:** 3
**Recommendation:** 5
**Confidence:** 2

**Main Review:**

Strengths：
1, I think the posterior noise-transition matrix and information fusion (IF) approach are new and interesting. Theoretical analyses are also provided to prove the robustness of such a transition matrix.

2, The paper is well written and organized. The authors did a good literature review on the previous methods.

Weaknesses:

1: In Theorem 3.5, authors assume that $e_T(x)$ and $e_W(x)$ are independent. I think it is impractical. By the definition of $e_T(x)$ and $e_W(x)$, these two terms may be strongly dependent. Thus Theorem 3.5 can not well explain the experiments.

2: From experiments in Table 1, PTM does not show much performance gain compared to PTD. Thus the applicability of the posterior transition matrix may be limited.
Besides, it is not clear how PTM and IF behave on the dataset with more categories such as CIFAR100.

3:  Many methods have been proposed in the literature to deal with IDN (A1, A2, A3, A4, A5). However, the authors only compare PTD in the experiments. I think more comparisons are needed to further validate the effectiveness of IF.


A1: Learning with bounded instance- and label-dependent label noise. ICML 2020

A2: Part-dependent label noise: Towards instance-dependent label noise. NeurlPS 2020

A3: Confidence scores make instance-dependent label-noise learning possible. ICML2021

A4: Learning with feature-dependent label noise: A progressive approach. ICLR2021

A5: Learning with instance-dependent label noise: A sample sieve approach. ICLR2021

**Summary Of The Paper:**

This paper provides new insight on estimating noise-transition matrix in the setting of instance-dependent label noise. Specifically, the authors observe that the traditional noise-transition matrix cannot well bridge the gap between underlying distribution and empirical distribution. Inspired by this observation, the authors propose to estimate the posterior noise-transition matrix. Further, authors use information fusion (IF) to linearly combine noise-transition matrix and posterior noise-transition matrix to obtain more accurate transition matrix estimation with theoretical guarantees. Experiments are conducted on CIFAR-10, SVHN, F-MNIST, and real-world dataset Clothing-1M.

**Summary Of The Review:**

Overall, the quality of this paper is good. The proposed approach is supported by the theorem. The experiment shows that information fusion (IF) outperforms PTD (Part-dependent transition matrix) by a large margin when the noise rate is high.

---

> ### Author Response · Authors · 2021-11-18
> **Response to Reviewer 8i2r [part 1 independent assumption]**
>
> We really appreciate the reviewer for the agreements on the importantness of our research problem and the significant technical contributions.  We would like to provide more explanations to address the reviewer’s concerns.
>
> **Q1: The assumption in Theorem 3.5, $e_T(x)$ and $e_W(x)$ are independent, is impractical. Theorem 3.5 can not well explain the experiments.**
>
> A1: Thanks for your insightful comment and careful review. We extend Theorem 3.5 to prove that IF can achieve lower estimation error **for general $e_T(x)$ and $e_W(x)$, either correlated or independent.** Specifically, we adopt a similar proof sketch to calculate the minimal reconstruction error is lower than that for PTM and PTD. The high-level intuition that the IF can boost estimation accuracy is **adaptively choosing a better estimator for each instance.** Furthermore, we compare the transition matrix estimation error on CIFAR-10 in Figure 3. It is seen that the reconstruction error of our IF is consistently the lowest under the different noise ratios. These experimental results in Table 1 further validate the correctness of Theorem 3.5, even with correlated $e_T(x)$ and $e_W(x)$. Please see Theorem 3.5 and Appendix A.5 for more details.

---

> ### Author Response · Authors · 2021-11-18
> **(Continued) Response to Reviewer 8i2r [part 2 experiments]**
>
> **Q2: PTM does not show much performance gain compared to PTD in experiments, which limited the applicability of PTM. Besides, it is not clear how PTM and IF behave on the dataset with more categories such as CIFAR100.**
>
> A2: We partially agree with your comment: Although PTM shows comparable performance with NTM, the applicability potential of PTM is maximized by leveraging the proposed IF to comprehensively correct the transition matrix estimation.
>
> First, we would like to highlight that PTM outperforms NTM with a large margin in CIFAR10 with 50% noise ratio and SVHN with 10% noise ratio while performing closely in most cases. Theoretically, NTM can be only used for the impractical mismatch problem between the underlying clean and underlying noisy data distribution. Compared with NTM, the superiority of PTM is brought by considering the gap between the empirical clean and empirical noisy data distribution, which is the actual distribution mismatch problem during model training. In Theorem 3.1, we theoretically show that PTM not only bridges the underlying clean and underlying noisy data distribution gap but also for the empirical counterpart for anchor points. In other words, PTM has more potential to tackle the label noise problem, which explains why PTM generally outperforms PTD in the low noise ratios. Considering the high noise ratios, as explained at the beginning of Section 3.3, the reason for the deterioration of the performance of PTM is the poor quality of PTM estimation.
>
> Second, to further reduce the estimation error PTM, we propose IF in Section 3.3 via linearly combining the PTM estimation and NTM estimation. We empirically and theoretically justify the design of IF in correcting the posterior transition matrix estimation. As shown in Table 1, our IF based on the proposed concept of PTM consistently deliver superior results in all the considered datasets and noise ratios. This empirically demonstrates the general applicability of PTM concept in the label denoising domain, which is firstly proposed in this paper.
>
> In summary, IF is empirically and theoretically guaranteed to provide low estimation error and generally good performance by incorporating PTM and NTM.
>
> **Q3: Additional baselines about recent literature dealing with IDN would benefit the experiments.**
>
> A3: Thanks for your constructive comment. Following most of the previous efforts at label noise correction, we mainly compare with the transition matrix-based approaches, where PTD [A2] is the highly related and SOTA baseline. To address your concerns, we have included more other baseline studies for CIFAR10, including instance-level forward correction (ILFC) [A3], progressive label correction (PLC) [A4], confidence regularized sample sieve (CORES) [A5]. As for [A1], we can only find Matlab code online. Due to the time limit for rebuttal, we do not reproduce the [A1] baseline and we believe that current results are sufficient to validate the effectiveness of our method. The performance of all methods is shown in the following table. It could be observed that our IF approach consistently deliver superior results, which further validates the effectiveness of leveraging the posterior transition matrix to correct noisy label.
>
> | Methods | IDN-10% | IDN-20% | IDN-30% | IDN-40% | IDN-50% |
> |   :----: |    :---: |    :---: |    :---: |    :---: |    :---: |
> |PLC|66.38 ± 0.61|60.42 ± 0.65|51.60 ± 0.58|42.77 ± 0.50|36.00 ± 1.73|
> |CORES|67.39 ± 0.44|60.85 ± 0.38|51.47 ± 0.80|43.96 ± 0.68|34.85 ± 1.66|
> |ILFC|80.22 ± 0.33|74.46 ± 0.09|73.27 ± 0.24|57.00 ± 4.07|36.27 ± 0.69|
> |PTD-F| 79.77 ± 0.91| 74.96 ± 0.71| 70.68 ± 0.81| 61.92 ± 1.59| 45.34 ± 4.67|
> |PTD-F-V| 80.08 ± 0.86| 74.67 ± 0.36| 71.66 ± 1.05| 62.45 ± 1.73| 46.16 ± 4.48|
> |PTM-F|78.16 ± 0.36|74.81 ± 0.81|70.03 ± 0.38|63.48 ± 0.38| 51.03 ± 3.08|
> |PTM-F-V|78.58 ± 0.31|75.06 ± 0.56|69.83 ± 0.58|62.69 ± 0.69|50.53 ± 3.41 |
> |IF-F| 80.92 ± 0.28| **79.58 ± 0.52**| 74.34 ± 0.86| **68.21 ± 2.21**| 50.07 ± 3.95|
> |IF-F-V| **80.94 ± 0.43**| 79.54 ± 0.45| **74.67 ± 0.92**| 68.03 ± 2.90| **52.34 ± 1.31**|
>
> [A1]Cheng, Jiacheng, Tongliang Liu, Kotagiri Ramamohanarao, and Dacheng Tao. Learning with bounded instance- and label-dependent label noise. ICML 2020
>
> [A2]: Xia, Xiaobo, Tongliang Liu, Bo Han, Nannan Wang, Mingming Gong, Haifeng Liu, Gang Niu, Dacheng Tao, and Masashi Sugiyama. Part-dependent label noise: Towards instance-dependent label noise. NeurlPS 2020
>
> [A3] Antonin Berthon, Bo Han, Tongliang Liu, Gang Niu, and Masashi Sugiyama. Confidence scores make instance-dependent label-noise learning possible. ICML, 2021
>
> [A4]Zhang, Yikai, Songzhu Zheng, Pengxiang Wu, Mayank Goswami, and Chao Chen. "Learning with feature-dependent label noise: A progressive approach." ICLR, 2021.
>
> [A5]Cheng, Hao, Zhaowei Zhu, Xingyu Li, Yifei Gong, Xing Sun, and Yang Liu. "Learning with instance-dependent label noise: A sample sieve approach." ICLR, 2021.

---

> > ### Comment · Reviewer_8i2r · 2021-11-22
> > **Response to authors**
> >
> > Thanks for your detailed reply. However, my major concerns about the experiments still remain.
> >
> > - As mentioned in my initial review, authors should perform CIFAR100 experiments to test how PTM and IF behave on the dataset with more categories. However, this part of the experiments is still missing in the current submission.
> >
> > - Why the performances of PLC and CORES are very low in your setting (even worse than CE baseline). This is very curious and counter-intuitive since these methods are also designed to deal with instance-dependent label noise. I think more fair comparisons are needed to verify the effectiveness of IF.

---

### Official Review · Reviewer_oWz1 · 2021-11-04

**Correctness:** 3
**Technical Novelty And Significance:** 4
**Empirical Novelty And Significance:** 4
**Recommendation:** 5
**Confidence:** 4

**Main Review:**

The paper proposes an interesting direction of estimating "instance-based" noise transition matrix, rather than estimating a class-based transition matrix. The authors first provide a brief introduction of noise transition matrices in the existing work, and motivate the need for estimating an instance-based transition matrix.

The presentation and the writing of the actual method can be improved to make it easier for the reader to understand different steps of the pipeline. For example, the interaction of different components (the network, PTM estimation IF correction etc. ) is not very clear, and an overview figure can be used to visualize their interaction. Figure 1 is not clear and does not help reader understand the interaction of different components.

My biggest concern is the assumption of "good" network outputs to estimate the PTM. The network is known to overfit the noisy labels, and this assumption may not always work well, especially if the noise ratio is high. For example, what happens when the noise ratio is above 50%. Existing work usually report noise ratio up to 80%-90% on CIFAR, and it would be interesting to see how PTM behaves under such setting.

The experiments would also benefit from additional baselines. For example, Label smoothing (Szegedy et al., 2016) adds a uniform distribution to the assigned label. Mixup (Zhang et al., 2018) creates a combination of labels. These methods do not correct the label noise through an instance-based transition matrix like this paper, however the way the "corrupt" the assigned label adds a regularization effect and help denoise the noisy labels.

The Clothing1M results in Table 2 are missing better-performing comparisons, such as Divide-Mix (Li et al., 2019), ELR+ (Liu et al., 2020) etc.


Christian Szegedy, Vincent Vanhoucke, Sergey Ioffe, Jon Shlens, and Zbigniew Wojna. Rethinking the inception architecture for computer vision. In CVPR, pp. 2818–2826, 2016.

Hongyi Zhang, Moustapha Cisse, Yann N Dauphin, and David Lopez-Paz. mixup: Beyond empirical risk minimization. ICLR, 2018.

Junnan Li, Richard Socher, and Steven CH Hoi. Dividemix: Learning with noisy labels as semisupervised learning. In ICLR, 2019.

Sheng Liu, Jonathan Niles-Weed, Narges Razavian, and Carlos Fernandez-Granda. Early-learning regularization prevents memorization of noisy labels. NeurIPS, 2020.

**Summary Of The Paper:**

This paper proposes to estimate an input-dependent noise transition matrix (PTM). The authors make a strong assumption that the neural network output approximates the underlying clean probability and estimate the PTM by extracting the output of each training sample using the network. To enforce the network to have better probabilities, PTM estimation is applied iteratively and warm-up is used during the training. The experiments are conducted in CIFAR-10 with different noise ratio, as well as a real world dataset, i.e. Clothing-1M.


**Summary Of The Review:**

The paper proposes an interesting and original idea. However, I think a few important experiments missing (higher noise ratio, additional baselines such as label smoothing and mixup), and the Table 2 needs to be updated with more recent work. I would be willing to upgrade my evaluation if the authors address my concerns during the rebuttal.

---

> ### Author Response · Authors · 2021-11-18
> **Response to Reviewer oWz1 (Part 1 system overview)**
>
> We really appreciate the reviewer for the recognition of the originality and interestingness of our work, and the constructive comments to improve this work. We would like to provide more explanations to address your concerns one by one.
>
> **Q1: The writing of the actual method can be improved. The overview figure can be used to visualize the interactions between the network, PTM estimation and IF correction.**
>
> A1: As required by the reviewer, we have added the overview figure and the systematic introduction on page 4 of the revised version, which is highlighted in blue. Specifically, our framework of information fusion (IF) consists of three key modules, including NTM and PTM estimation (see Section 3.2), IF correction (see Section 3.3), and posterior loss function (see Section 3.1). First, while NTM estimation is adopted to correct the underlying clean and underlying noisy data distribution match before neural network training [1], the actual distribution gap falls in empirical clean and empirical noisy distribution mismatch. Therefore, we propose PTM to tackle the empirical clean and empirical noisy distribution mismatch problem by comprehensively considering the neural network prediction results and observed noisy label. Second, the IF leverages the linear combination between NTM (fixed during training) and PTM (iteratively update during training) to obtain a more accurate estimated transition matrix. At last but not least, the posterior loss correction is adopted to tackle instance-dependent label noise.
>
> Furthermore, we would like to point out that Figure 1 is paramount to explain the shortcoming of NTM and understand our research problem's importance in depth. NTM works towards bridging the gap between underlying clean and underlying noisy data distributions, while the actual distribution mismatch problem during training appears **between the empirical clean and empirical noisy data distributions**. As shown by the intuitive example in Figure 1 (the upper part), NTM fails to model the realistic distribution gap. To tackle such a problem, we proposed PTM to describe label noisy characterization given the observed noisy label (posterior information). As illustrated by the intuitive example in Figure 1 (the lower part), PTM can discriminate label corruption with different noisy label observations even if generated from the same label noise. We also justify PTM via demonstrating the statistical consistent classifier using PTM and posterior loss correction method.
>
> In summary, we have illustrated the framework overview in Figure 2 (in the revised version) to make our technical part more clear and re-explained the key role of Figure 1 to help understand our research motivation. We will keep improving the writing during the rebuttal if you could kindly let us know more other concerns.
>
> [1] Xiaobo Xia, Tongliang Liu, Bo Han, Nannan Wang, Mingming Gong, Haifeng Liu, Gang Niu, Dacheng Tao, Masashi Sugiyama. Part-dependent label noise: Towards instance-dependent label noise. NeurlPS 2020

---

> ### Author Response · Authors · 2021-11-18
> **(Continued) Response to Reviewer oWz1 [Part 2 methodology and experiments]**
>
> **Q2: The biggest concern is on the assumption of "good" network outputs for PTM estimation. This assumption may not always work well, especially for the high noise ratio, since the network is known to overfit the noisy label.**
>
> A2: We totally agree with you that the "good" network outputs do not always work well for PTM estimation, especially in the high noise ratio case. At the beginning of Section 3.3 on Page 6, we have pointed out the condition that neural network approximates clean label is quite strong and limited, under which PTM estimation error could be large for large IDN rate. Motivated by such limitation, we thus propose IF to correct the estimated transition matrix via linearly combining PTM estimation and NTM estimation. As justified in Theorem 3.5, we theoretically demonstrate that our IF could achieve lower transition matrix estimation error compared with PTD and PTM. To further empirically demonstrate the effectiveness of IF, we compare the transition matrix estimation error for PTD, PTM, and IF in CIFAR10 dataset, where IF can achieve the lowest estimation error for all noise ratios.
>
> By comprehensively considering the high noise ratios of 70%, 80%, and 90% on dataset CIFAR10, we compare PTD, PTM, and IF in terms of classification accuracy in the following table. We have the following two findings: (1) Comparing with PTD at the noise ratios of 80% and 90%, our PTM could still outperform PTD. Although the network output is not perfect, the performance deterioration of PTD is still higher than PTM. (2) Our IF consistently delivers the most superior performance no matter under the low or high noise. That is because IF adaptively and linearly combines NTM and PTM in instance level and thus achieves more accurate estimation.
>
> In summary, the experiment results validate the effectiveness of our IF method, which is consistent with our theoretical and empirical analysis in the manuscript. We have provided more experiments on the high noise ratios in Appendix C.2 of the revised version.
>
> | Methods | IDN-70% | IDN-80% | IDN-90% |
> |   :----: |    :---: |    :---: |    :---: |
> | PTD-F | 20.25 ± 1.56|13.48 ± 0.81|9.22 ± 0.23|
> | PTD-F-V |20.35 ± 1.56 |13.58 ± 0.80|9.44 ± 0.24|
> | PTM-F | 16.79 ± 3.86|14.13 ± 0.34|10.34 ± 0.71|
> | PTM-F-V |18.95 ± 2.89 |13.89 ± 0.42|10.57 ± 0.82|
> | IF-F | **21.26 ± 1.46**|15.78 ± 1.65|10.58 ± 0.53|
> | IF-F-V |21.09 ± 0.45 |**16.72 ± 0.61**|**10.86 ± 0.40**|
>
> **Q3: Additional baselines are needed in the experiments, such as Label smoothing, Mixup, Dividemix, ELR.**
>
> A3: Thanks for your constructive comment. Following most of the previous efforts at label noise correction, we mainly compare with the transition matrix-based baselines. To address your concerns, we have included more other baseline studies, including label smoothing (LS) [2], early-learning regularization (ELR) [3], and instance-level forward correction (ILFC) [4]. As shown in the following table, considering the general noise ratios on CIFAR10, we could observe that our method consistently achieves state-of-the-art performance. We also add the performance of ELR for real-world dataset Clothing1M in Table. 2 of the revised manuscript.
>
> In the revised version, we will make the baseline comparison more complete by running on other benchmark datasets. Please see Appendix C.3 for more details.
>
> | Methods | IDN-10% | IDN-20% | IDN-30% | IDN-40% | IDN-50% |
> |   :----: |    :---: |    :---: |    :---: |    :---: |    :---: |
> |LS| 63.19 ± 0.25| 57.67 ± 0.10| 49.50 ± 0.17| 41.57 ± 1.85| 34.89 ± 2.47|
> |ELR| 66.20 ± 0.09| 60.58 ± 0.75| 51.65 ± 0.33| 43.83 ± 1.61| 34.44 ± 0.12|
> |ILFC|80.22 ± 0.33|74.46 ± 0.09|73.27 ± 0.24|57.00 ± 4.07|36.27 ± 0.69|
> |PTD-F| 79.77 ± 0.91| 74.96 ± 0.71| 70.68 ± 0.81| 61.92 ± 1.59| 45.34 ± 4.67|
> |PTD-F-V| 80.08 ± 0.86| 74.67 ± 0.36| 71.66 ± 1.05| 62.45 ± 1.73| 46.16 ± 4.48|
> |PTM-F|78.16 ± 0.36|74.81 ± 0.81|70.03 ± 0.38|63.48 ± 0.38| 51.03 ± 3.08|
> |PTM-F-V|78.58 ± 0.31|75.06 ± 0.56|69.83 ± 0.58|62.69 ± 0.69|50.53 ± 3.41 |
> |IF-F| 80.92 ± 0.28| **79.58 ± 0.52**| 74.34 ± 0.86| **68.21 ± 2.21**| 50.07 ± 3.95|
> |IF-F-V| **80.94 ± 0.43**| 79.54 ± 0.45| **74.67 ± 0.92**| 68.03 ± 2.90| **52.34 ± 1.31**|
>
>
> [2] Christian Szegedy, Vincent Vanhoucke, Sergey Ioffe, Jon Shlens, and Zbigniew Wojna. Rethinking the inception architecture for computer vision. In CVPR, pp. 2818–2826, 2016.
>
> [3] Sheng Liu, Jonathan Niles-Weed, Narges Razavian, and Carlos Fernandez-Granda. Early-learning regularization prevents memorization of noisy labels. NeurIPS, 2020.
>
> [4] Antonin Berthon, Bo Han, Tongliang Liu, Gang Niu, and Masashi Sugiyama. Confidence scores make instance-dependent label-noise learning possible. ICML2021

---

### Decision · Program_Chairs · 2022-01-20

**Decision:**

Accept (Poster)

**Comment:**

To tackle the problem of classification under input-dependent noise, the authors proposed the posterior transition matrix (PTM) to achieve statistically consistent classification. Specifically the information fusion approach was developed to fine-tune the noise transition matrix. Experiments demonstrated the effectiveness of the proposed approach.

I would like to thank the authors for the detailed feedback to the initial reviews and also further feedback to the reviewers' additional questions. Many concerns were clarified by the feedback, and the additional experiments still demonstrate the effectiveness of the proposed method.

The issue of data augmentation still remains, which should be at least experimentally investigated,
but the contribution of the current manuscript is still valuable to be presented as ICLR2022.

---

> ### Public Comment · ~Xinshao_Wang1 · 2022-07-10
> **An Open Question on whether clean or noisy validation set for ML/DL researchers caring about label noise**
>
> The authors mentioned: **The real-world dataset Clothing1M has 1M images with real-world noisy labels and 10k images with clean labels for testing. In the experiments, we leave out 10% of the noisy training samples as a noisy validation set for model selection.**
>
> However, please check the following discussions about **the flaw of this specific experimental setting**. To highlight, **I only would like to discuss this experimental setting, and I am not commenting on the technical quality of this paper.**
>
> * https://github.com/XinshaoAmosWang/Improving-Mean-Absolute-Error-against-CCE#open-reviews-and-discussion.
> * https://www.reddit.com/r/MachineLearning/comments/htgucz/r_when_talking_about_robustnessregularisation_our/?utm_source=share&utm_medium=web2x&context=3
> * https://www.reddit.com/r/MachineLearning/comments/hmt9ds/r_we_really_need_to_rethink_robust_losses_and/?utm_source=share&utm_medium=web2x&context=3
>
> ### Very nice discussions in reddit about "**A validation set serves a similar rule as a testing set. Both have to be clean.**"
> Link: https://www.reddit.com/r/MachineLearning/comments/nsh6uu/r_cvpr_2021progressive_self_label_correction/h10ga71/?utm_source=share&utm_medium=web2x&context=3
>
> Thanks.

---

> ### Public Comment · ~Xinshao_Wang1 · 2022-07-10
> **Discussion: Minimizing risks on the noisy validation set is NOT asymptotically equal to minimizing risk on the clean data because the noise rate is usually unknown!**
>
> https://openreview.net/forum?id=xENf4QUL4LW&noteId=C2eCHs2k6CM